


# **Bromine speciation and partitioning in slab-derived fluids and**
# **melts: Implications for halogens recycling in subduction zones**
Marion Louvel[1,2], Carmen Sanchez-Valle[2], Wim J. Malfait[3], Gleb S. Pokrovski[4], Camelia N.
Borca[5] and Daniel Grolimund[5]
[1] School of Earth Sciences, Bristol University, UK- BS81RJ, Bristol, United-Kingdom
[2] Institute for Mineralogy, WW-Universität Münster, D-48149, Münster, Germany
[3] Swiss Federal Laboratories for Materials Science and Technology EMPA, CH-8600, Dubendorf, Switzerland
[4] Groupe Métallogénie Expérimentale, Géosciences Environnement Toulouse (GET), OMP-CNRS-IRD-University
of Toulouse, 14 Avenue Edouard Belin 31400 Toulouse, France
[5] Swiss Light Source, Paul Scherrer Institute, CH-5232, Villigen, Switzerland
*Correspondence to:* Marion Louvel (louvel@uni-muenster.de)
**Abstract.** Understanding the behavior of halogens (Cl, Br, and I) in subduction zones is critical to constrain the
recycling of trace elements and metals, and to quantify the halogen fluxes to the atmosphere *via* volcanic degassing.
Here, the partitioning of bromine between coexisting aqueous fluids and hydrous granitic melts and its speciation in
slab-derived fluids have been investigated *in situ* up to 840 °C and 2.2 GPa by X-ray fluorescence (SXRF) and
absorption (XANES and EXAFS) spectroscopy in hydrothermal diamond-anvil cells. The partition coefficients
$D^{f/m}_{Br}$ range from 15.3 ±1.0 to 2.0 ±0.1, indicating the preferential uptake of Br by aqueous fluids at all investigated
conditions. EXAFS analysis further evidences a gradual evolution of Br speciation from hydrated Br ions
$[Br(H_2O)_6]^-$ in slab dehydration fluids to more complex structures invoving both Na ions and water molecules,
$[BrNa_x(H_2O)_y]$, in hydrous silicate melts and supercritical fluids released at greather depth (> 200 km). In dense
fluids containing 60 wt% dissolved alkali-silicates and in hydrous $Na_2Si_2O_5$ melts (10 wt% H2O), Br is found in a
"salt-like" structure involving 6 nearest Na ions and several next-nearest O neighbors that are either from water
molecules or the tetrahedral silicate network. Bromine (and likely chlorine and iodine) complexation with alkalis is
thus an efficient mechanism for the mobilization and transport of halogens by hydrous silicate melts and
supercritical fluids, which can carry high amounts of Br, up to the 1000 ppm level. Overall, our results suggest that
both shallow dehydration fluids and deeper silicate-bearing fluids efficiently remove halogens from the slab in the
sub-arc region, thus controling an efficient recycling of halogens in subduction zones.
***Keywords:*** Bromine, volatile elements, fluxes, speciation, partitioning, slab fluids, halogens, geochemical cycle,
subduction zones.





**1 Introduction**

The fluxes of volatile elements (water, carbon, sulfur, and halogens) in subduction zones

play a critical role in the Earth's chemical evolution by controlling the transfer of slab
components to the mantle wedge, the volcanic arc and, ultimately, to the atmosphere. Although
halogens (F, Cl, Br and I) are rather minor volatiles compared to $H_2O$ and $CO_2$, their effect on
the physical and chemical properties of slab-derived fluids and arc magmas (*e.g.* phase
equilibria, viscosity, density), as well as their ability to complex with trace elements and metals
makes them key players in the chemical transfer in subduction zones (Zellmer et al., 2015;
Barnes et al., 2018). Furthermore, their emission to the troposphere and stratosphere at volcanic
arc centres may have significant environmental impact, including 'forced' ozone depletion by Br
(Bobrowski et al., 2003; von Glasow et al., 2009; Kutterolf et al., 2013). Constraining halogens
cycle at subduction zones is thus crucial to assess their impact on the global atmospheric
chemistry and climate.

In the last decade, new developments in quantification technics on pore fluids and rocks

as well as in detection methods for halogens species in volcanic gases enabled better estimates of
halogen fluxes in subduction zones (Wallace, 2005; Pyle and Mather 2009; John et al., 2011;
Kendrick et al., 2013; Kendrick et al., 2015; Chavrit et al., 2016; Barnes et al., 2018).
Comparison of the input from the subducted sediments, altered oceanic crust and serpentinized
oceanic lithosphere to the output along volcanic arcs points to a strong imbalance between
fluorine input and output, suggesting a significant amount of F may be recycled into the mantle
(Roberge et al., 2015; Grutzner et al., 2017). On the contrary, Cl, Br and I appear to be
efficiently recycled up to the surface, either through shallow loss of pore fluids to the fore-arc
region (Br and especially I) or deeper release upon slab dehydration (especially Cl and Br, and to



a lesser extent I) (Kendrick et al., 2018). Yet, the poor understanding of the transfer mechanisms
and recycling paths of halogens limits the development of integrative numerical models
constraining the role of fluids and halogens in the global cycling of elements in subduction zones
(Ikemoto and Iwamori, 2014; Kimura et al., 2016). There is for instance virtually no constraint
on the amounts of residual halogens that may be stored in the dehydrated slab or lost through
hidden hydrothermal activity and passive degassing in the continental crust. Similarly, current
knowledge of halogens solubility and speciation in fluids and melts is mostly limited to pressures
below 0.3 GPa (equivalent to ~10 km depth), which are relevant to volcanic degassing and ore
deposit formation in the shallow crust, but not to slab dehydration or melting beneath arcs at far
greater depth (Webster, 1990; Métrich and Rutherford, 1992; Webster, 1992; Bureau et al., 2000;
Signorelli and Carroll, 2002; Bureau and Métrich, 2003; Carroll, 2005; Evans et al., 2009;
Cadoux et al., 2018). Only recently Bureau et al. (2010, 2016) reported fluid-melt partition
coefficients for Br and I in the haplogranite-$H_2O$ system up to 1.7 GPa while Cochain et al.
(2015) investigated the speciation of Br in aqueous fluids or haplogranitic melts up to 7.6 GPa.
Nevertheless, the effect of fluid chemistry on the speciation and partitioning of halogens at high
pressures and temperatures (*P-T*) is unknown in subduction zones. To fill this gap, we combine
Synchrotron X-ray Fluorescence (SXRF) and X-ray Absorption Spectroscopy (XAS)
measurements in a hydrothermal diamond-anvil cell (HDAC) to investigate Br fluid-melt
partitioning and speciation in aqueous fluids and hydrous silicate melts that mimic the mobile
phases released by the slab at sub-arc depths (Manning, 2004; Frezzotti and Ferrando, 2015).
Bromine is employed here as an analog of chlorine amenable to SXRF and XAS studies through
the diamond window of the diamond anvil cell due to its higher absorption edge energy, 13.47
keV for bromine K-edge compared to 2.82 keV for chlorine K-edge (Sanchez-Valle, 2013).



Furthermore, among the halogens, bromine displays the closest behavior to chlorine in terms of
solubility, partitioning and speciation in silicate melts, at least at shallow depths (Bureau et al.,
2000, Bureau and Metrich, 2003; Wasik et al., 2005; Bureau et al., 2010; Cadoux et al., 2018).
Bromine therefore represents the best analog of Cl for *in-situ* studies at high pressure and high
temperature conditions. Our experimental results show gradual changes in the speciation of
bromine that reflect changes in fluid composition with depth and constrain the mechanisms
controlling the transfer of halogens from the slab to arc magmas.

**2. Methods**
**2.1. Synthesis and characterization of starting materials**

The speciation and fluid-melt partitioning experiments were conducted using 3 wt% NaBr

aqueous solutions and synthetic sodium disilicate (NS2: $Na_2Si_2O_5$) or haplogranite (Hpg) glasses
doped with 1 to 4 wt% Br as starting materials (Table 1). The 3 wt% NaBr aqueous solution was
freshly prepared from distilled de-ionized water and analytical grade NaBr powder, sealed in
tight containers and refrigerated until the experiments. The NS2 and haplogranite glasses were
synthesized in a piston-cylinder apparatus at 1200 °C and 0.5 GPa and 1.5 GPa, respectively,
following the method described in Louvel et al. (2013). Briefly, reagent grade powders of $SiO_2$
and $Na_2SiO_3$ were employed for the NS2 glasses whereas reagent grade $SiO_2$, $Al_2O_3$ and alkali-
carbonates, $K_2CO_3$ and $Na_2CO_3$, were mixed for the haplogranite glass synthesis. Bromine was
added as NaBr powder, and 3.3 wt% $H_2O$ was added as liquid water for the synthesis of the
haplogranite glass to ensure complete melting and homogeneization of the sample at run
conditions.

Major element composition, Si, Al, K and Na, and glass homogeneity were checked by

electron microprobe analyzer (EMPA) using a JEOL JXA-8200 microprobe (Table 1). Electron





microprobe analyses of Br are hindered by i) the high ionization potential for the K-lines of Br,
which results in low count rates; ii) the peak overlap between the L-lines of Br and the K-lines of
Al; and iii) the lack of matrix-matched standards. To overcome these limitations, the
concentration of Br in Hpg-Br2 glass sample was determined by Rutherford Backscattering
Spectroscopy (RBS). This technique provides absolute elemental concentrations and is
particularly appropriated for the quantification of heavy elements in a light matrix as it is the
case of Br in silicate glasses (Feldman and Mayer, 1986; Chu and Liu, 1996). The RBS analysis
yielded a Br concentration of $0.96 \pm 0.04$ wt% (Fig. S1 in Supplementary Material), which is
identical to the nominal Br concentration within analytical uncertainties. This well-characterized
sample was then used as a standard for Br analysis by EMPA and LA-ICPMS in the other glass
samples (Table 1 and Supplementary Material).

**2.2 Hydrothermal diamond anvil cell experiments**
All experiments were conducted in Bassett-type hydrothermal diamond-anvil cells
(HDAC, Bassett et al., 1993) widely used for *in-situ* X-ray fluorescence (SXRF) and absorption
(XAS) measurements on aqueous fluids and silicate melts up to 1000 °C and about 3 GPa (e.g.
Borchert et al., 2009; Louvel et al., 2013, 2014). The HDACs were mounted with a thinner
diamond (1.2 mm thick) on the detector side to reduce the X-ray path through the diamonds and
widen the collection angle of the XAS analysis (Sanchez-Valle et al., 2004). This configuration
allows i) reducing the attenuation of the fluorescence X-rays in the anvil, and ii) decreasing the
fluorescence background arising from the Compton/Rayleigh scattering in the thick diamond
anvils, hence increasing the overall quality of the analysis. The sample chamber, a 300 μm hole
drilled in a 250 μm rhenium gasket compressed between the two diamond anvils, was heated



externally with Mo wires wrapped around the tungsten carbide seats supporting the diamond
anvils. Temperature was measured to within 2 °C with K-type thermocouples attached to each
diamond-anvil, as close as possible to the sample chamber. The temperature gradient between
thermocouples and the sample chamber was calibrated for each HDAC prior to experiments
using the melting temperature at ambient pressure of S (115.4 °C), $NaNO_3$ (308 °C) and NaCl
(800.5 °C). Overall, they remain below 30-35 °C at the highest temperature reached. Pressure
was determined from the equation of state of the gold internal pressure standard (Jamieson et al.,
1982) whose X-ray diffraction pattern was measured during the experiment.

Partitioning experiments were conducted by loading the sample chamber with a piece of

Br-bearing haplogranite glass and either pure $H_2O$ or a 3 wt% NaBr solution (Fig. 1). For the
speciation studies by XAS, loadings included either Br aqueous solutions, or a piece of Br-
bearing NS2 or haplogranite glass loaded together with distilled de-ionized water. In all runs, a
pellet of a mixture of Au +$Al_2O_3$ powders was added to be used for pressure calibration (Louvel
et al., 2013; 2014). The volumetric proportions of glass and aqueous fluid in the different
loadings were adjusted by adding double-side polished glass pieces of known dimensions (Fig.
1). Upon heating, the haplogranite melt-aqueous fluid system followed the classical phase
transitions described in previous studies (Bureau and Keppler, 1999; Louvel et al., 2013), with
initial hydrous melting recorded between 550 and 700 °C (Fig. 1B) and complete miscibility
reached around 700-850 °C depending on the pressure (Fig. 1C). In contrast, the NS2-$H_2O$
system displays distinct and unusual phase relations in the investigated pressure-temperature
range (Fig. 1D-F): the NS2 glass first dissolves completely in the aqueous solution between 150
and 250 °C to produce a single fluid phase, or hydrosilicate liquid, containing 30 to 60 wt%





dissolved $Na_2O$ and $SiO_2$ solutes (Fig. 1E). Upon further heating between 500 and 750 °C, the
fluid unmixes into two phases, a hydrous melt and an aqueous fluid (Fig. 1F). The high
temperature immiscibility gap remains open up to the highest temperatures reached with the
HDAC (~ 800-900 °C), as previously observed for the haploandesite $Na_2Si_4O_9$-$Na_2(Si,Al)_4O_9$
join and the $K_2O$-$SiO_2$-$H_2O$ system (Mysen and Cody, 2004).

The composition of the high pressure fluids (wt% cations dissolved) and melts (wt% $H_2O$)

was determined from available solubility studies (Table 2) as follows. The water content of
haplogranite melts at equilibrium with aqueous fluids (Fig. 1B) was calculated from the water
solubility data for aluminosilicate melts reported by Mysen and Wheeler (2000) and extended to
our experimental conditions. The composition of the aqueous fluids in equilibrium with the
haplogranite melts (*i.e.,* total silicates content including $SiO_2$, $Al_2O_3$, $Na_2O$ and $K_2O$) was
calculated, by extrapolating up to the *P-T* conditions of our experiments, the solubility data
reported in the albite-$H_2O$ system between 0.20 and 0.84 GPa at 600 and 700 °C (Anderson and
Burnham, 1983). The composition of the hydrosilicate liquids in the NS2-$H_2O$ system was
determined from the initial volumetric proportions of the NS2 glass and the aqueous fluid loaded
in the compression chamber. The mass of the glass was calculated from the volume using a
density of 2.52(5) g/cm$^3$ (Yamashita et al., 2008) and that of the fluid determined from the
volume left in the compression chamber (Fig. 1D); 4) the amount of water dissolved in the
hydrous NS2 melt in equilibrium with the aqueous fluid at 700 °C and 0.4 GPa (Fig. 1F) was
calculated from water solubility data in sodium silicate melts reported by Mysen and Cody
(2004). The overall error in the calculated bulk compositions is within 10% of the total
concentration value.





**2.3 *In-situ* SXRF/XAS measurements and data analysis**
The SXRF and XAS measurements were performed at the MicroXAS beamline (X05-
LA) of the Swiss Light Source (SLS, Paul Scherrer Institute, Borca et al., 2009). Measurements
at the Br K-edge were conducted with an incident energy of 13.6 keV tuned by a Si(111) double
crystal monochromator and focused down to 5 x 8 (V×H) $\mu m^2$ size by a set of Rh-coated
Kirkpatrick-Baez mirrors. This configuration ensured a photon flux of $\sim 2 \times 10^{11}$ photons per
second at the measurements conditions. The intensity of the incident beam was monitored
throughout the experiments using an Ar-filled micro-ion-chamber placed between the
Kirkpatrick-Baez mirrors and the HDAC. Before measurements, temperature was stabilized for
about 30 min after each heating stage to ensure that chemical equilibrium was achieved inside
the cell (Louvel et al., 2014). In the case of coexisiting melts and fluid, careful attention was paid
that measurements were performed when the melt globule was stationary and bridging both
diamonds (Fig. 1F). This configuration ensured that data collection was performed in a pure
phase (either fluid or melt) without contamination of the SXRF and XAS signals by the
coexisting phase to. SXRF and XAS spectra were collected in fluorescence mode in a forward
scattering geometry with an energy dispersive single-element Si-detector (Ketek®, 139 eV
resolution at Mn-K$_{\alpha}$ = 5.89 keV) set at 22° from the incident beam in the horizontal plane
(Sanchez-Valle et al., 2003; Louvel et al., 2013; 2014). Angle-dispersive X-ray diffraction
spectra were collected on the gold pressure standard before and after XAS/SXRF measurements
using a high-resolution CCD camera set in transmission geometry. A microscope equipped with
a video camera was used to monitor the compression chamber during the heating and cooling
cycles (Fig. 1).





2D-SXRF maps were collected across the sample chamber to qualitatively monitor the
distribution of Br between coexisting aqueous fluids and haplogranite melts (Fig. 2). Then, a
minimum of three fluorescence spectra was collected in each phase to further determine the Br
fluid-melt partition coefficients $D_{Br}^{f/m}$ at each pressure-temperature condition (Table 2).
Counting times were set to 100 or 300 s, depending of the intensity of the signal. The fluid-melt
partition coefficients $D_{Br}^{f/m}$ at each pressure-tempeature were derived from the integrated
intensities of the Br fluorescence emission lines recorded in the fluid and melt, $I_f$ and $I_m$, after
normalization to the incident beam intensity and counting times, and background removal with
the Peakfit v4.12 software (SeaSolve Software-USA), following the method described in Louvel
et al. (2014). This method relies on the fixed geometry of the HDAC set-up and takes into
account the different composition, density (ϱ) and effective transmission (T) of the aqueous fluid
and melt to normalize the fluorescence signal and calculates $D_{Br}^{f/m}$ with an uncertainty below
10% according to the equation:

$$D^{f/m} = \frac{I_f}{I_m} \cdot \frac{T_m}{T_f} \cdot \frac{\rho_m}{\rho_f}$$    (1)
Correction parameters are provided in Table 2 and additional details for density and effective
transmission calculations are found in Louvel et al. (2014).

XAS measurements were conducted on 3 wt% NaBr aqueous solution, 'solute-poor' fluids
equilibrated with hydrous haplogranite melt (Fig. 1B), hydrosilicate liquids containing different
amounts of dissolved NS2 (Fig. 1E) and hydrous NS2 melt (Fig. 1F). XAS analyses on the
haplogranite melt were precluded by the lower Br concentration of these melts (probably < 1000





ppm). For each composition, 3 to 5 XAS spectra were collected with counting times of 1 second
per point in the pre-edge region to 3 seconds in the XANES and EXAFS regions. The
contribution of Bragg reflections arising from the diamond anvils was avoided in the energy
range of interest by changing the orientation of the diamond anvil cell by 0.5 to 1° with respect
to the incident X-ray beam direction (Bassett et al., 2000). The edge position was calibrated
using a pellet of NaBr powder and no significant drift of the energy was observed during
measurements. XAS spectra were also collected at ambient conditions on ~ 200 x 200 $\mu m^2$
double-side polished section of the NS2 and haplogranite glasses.

Data reduction was performed using the Athena and Artemis packages (Ravel and

Newville, 2005) based on the IFEFFIT program (Newville, 2001). Averaged experimental
spectra were normalized to the absorption edge height and background removed using the
automatic background subtraction routine AUTOBK included in the Athena software. To
minimize the contribution of features at distances below the atom-atom contact distance, the $R_{bkg}$
parameter, which represents the minimum distance for which information is provided by the
signal, was set to 1.3 Å. For all spectra, the absorption energy $E_0$ was set to 13.474 keV, which
corresponds to the maximum of the first derivative of the absorption edge. Based on previous
studies of Br and Cl speciation in aqueous solutions and silicate glasses (Ayala et al., 2002;
D'Angelo et al., 1993; Evans et al., 2008; Ferlat et al., 2001; McKeown et al., 2011; Ramos et
al., 2000; Sandland et al., 2004; Stebbins and Du, 2002), our EXAFS analysis includes the Br-O
and Br-Na scattering paths as end-members to describe the evolution of the local structure
around Br from the high P-T fluids to the hydrous melts and silicate glasses. Although Na cannot
be easily distinguished from Al or Si by EXAFS at our spectral resolution, the presence of
network cations in the nearest coordination shell of Br is deemed unlikely, as previously shown





for Cl by MAS-NMR and XAS studies (Evans et al., 2008; McKeown et al., 2011; Sandland et
al., 2004; Stebbins and Du, 2002). The theoretical back-scattering amplitudes F(k), mean free-
paths $\lambda(k)$ and phase-shift functions $\phi(k)$ for these paths were calculated with the FEFF6.0 *ab*
*initio* code (Mustre de Leon et al., 1991) using an aqueous Br ion $[Br(H_2O)_6]^-$ with mean Br-O
distance of 3.37 Å and the NaBr salt crystallographic structure with Br-Na distance of 2.98 Å
(Deshpande, 1961; Makino, 1995). Multiple scattering within a linear Br···H-O cluster was also
included to model the hydration shell around Br, with the H-O distance fixed to 1.0 Å
(Silvestrelli and Parrinello, 1999; Soper and Benmore, 2008). The $\chi(k)$ EXAFS function were
Fourier filtered over the 1.5 to 6 $Å^{-1}$ k-range for most spectra. For all samples, modelling of the
EXAFS oscillations was performed using 4 variables: average coordination number (N), distance
to nearest neighbor (R), Debye-Waller factor $\sigma^2$, and the energy shift $\Delta E$. The amplitude
reduction factor $S_0^2$ was set to 1 based on previous fits of aqueous NaBr, KBr and $GaBr_3$
solutions (Da Silva et al., 2009; Ferlat et al., 2002). All fits were performed simultaneously with
k-weighting of 1, 2 and 3 in order to decrease correlations between N and $\sigma^2$, and R and $\Delta E$
(Pokrovski et al., 2009a,b). The multi-electronic excitations (MEE) at 34 and 90 ($\pm$ 0.05) eV
above the Br K-edge (D'Angelo et al., 1993) were neglected as they did not significantly
contribute to the EXAFS spectra. The variation of $\Delta E$ values between different fitted samples
was less than $\pm$ 4 eV, further confirming the validity of the fitting procedure and the accuracy of
the derived interatomic distances.

**3 RESULTS AND DISCUSSION**
**3.1 Bromine partition coefficients in the haplogranite-fluid system**



The distribution of Br between aqueous fluids and silicate melts at high P-T conditions
has been constrained by measuring fluid-melt partition coefficients $D_{Br}^{f/m}$ from 592 to 840 °C
and 0.2 to 1.7 GPa in four experimental runs. For all investigated conditions, the $D_{Br}^{f/m}$ values
are always higher than 1 (Table 2), confirming the preferential partitioning of Br into the fluid
phase, which is also qualitatively evident from the *in-situ* Br distribution maps reported in Figure
2. $D_{Br}^{f/m}$ values vary between 2.0 ±0.1 and 15.3 ±1.0, and fall within the range reported in a
previous HDAC study by Bureau et al. (2010) at similar *P-T* conditions (Fig. 3). The $D_{Br}^{f/m}$
values are also found to decrease towards unity with increasing *P-T*, as the compositional and
structural differences between aqueous fluid and hydrous melt vanish when approaching the
critical conditions for the haplogranite-$H_2O$ system (Bureau and Keppler, 1999). Overall, an
important finding here is that the $D_{Br}^{f/m}$ values remain relatively small under high *P-T*
conditions. This observation suggests that hydrous melts have a capacity comparable to fluids to
carry Br, at 100s to 1000s ppm concentrations under high *P-T* conditions and may thus
contribute to the efficient transport and recycling of Br to the mantle wedge and volcanic arcs.

At low pressure conditions relevant to fore-arc or crustal processes, our *in-situ* partition
coefficients are, however, slightly lower than those obtained from quenched experiments (Fig.
3). For instance, Bureau et al. (2000) and Cadoux et al. (2018) reported average $D_{Br}^{f/m}$ of 17.5
±0.6 and 20.2 ±1.2 for albitic and rhyodacitic melts at 900 °C and 0.2 GPa, while we found
$D_{Br}^{f/m}$ of 4.8 ±0.3 at 800 °C and 0.2 GPa. We also note that the minimum $D_{Br}^{f/m}$ from Cadoux et
al. is of 8.6 ±2.2, even closer to our *in-situ* value. These rather small discrepancies may underline
issues with pressure determination at low pressure conditions in the HDAC, the quantification of



Br by mass balance techniques in Bureau et al. and Cadoux et al. (i.e, salt precipitates), or
artifacts of the quench method resulting in the loss of Br to the aqueous phase upon cooling.
Furthermore, slight differences in the melt composition and structure could also result in
different Br speciation (Louvel et al., 2020), favoring or not the incorporation of Br in the silicate
melt. Under these $P$ conditions and similarly low concentration, the available experimental $D_{Br^\pm}^{f/m}$
for silicic melts (Bureau et al., 2000; Cadoux et al., 2018; *this study*) are comparable to those
reported for Cl. Indeed, fluid-melt partition coefficients from experiments with low Cl
concentration (< 1m Cl) range between ~3 and 20 for phonolitic to rhyolitic composition
(Chevychelov et al., 2008; Webster and Holloway, 1988; see Dolejs and Zajacz, 2018 for a
review), further confirming the similar behavior of Br and Cl under magmatic conditions.

**3.2 Speciation of bromine in aqueous fluids and silicate melts**
**3.2.1 Aqueous solutions and silicate glasses at room conditions**
The XANES and EXAFS spectra collected at ambient conditions from the 3 wt%NaBr
aqueous solution and Br-bearing silicate glasses are reported respectively in Fig.4 and Fig.5,
together with data for a KBr aqueous solution from Ferlat et al. (2002). These spectra were
employed to validate the theoretical backscattering amplitude and phase shift functions for Br-O
and Br-Na scattering paths used in EXAFS modeling. The XANES spectrum of the 3 wt% NaBr
aqueous solution is characterized by an absorption edge at 13.474 keV and a white line that
peaks at 13.478 keV (Fig. 4). It displays close similarities to that of the KBr aqueous solution
from Ferlat et al. (2002) and overall resembles other alkali bromide aqueous solutions found in
the literature (Wallen et al., 1997; Ferlat et al., 2001; Evans et al., 2007). The EXAFS spectra
from the KBr and NaBr aqueous solutions are accurately modeled with a hydration shell of 5.7 ±





0.8 and 5.9 ± 0.7 water molecules ($N_{Br..H-O}$) at a Br-O distance of 3.30 ± 0.03 and 3.37 ± 0.04 Å,
respectively (Table 3). Note that multiple-scattering paths from the linear Br···H-O cluster are
needed to accurately reproduce the experimental data; when only Br-O interactions are
considered, the model fails to reproduce the amplitude of the EXAFS oscillations unless an
unrealistic hydration shell with ~12 $H_2O$ molecules is adopted. The structural parameters fitted
for the KBr aqueous solution from Ferlat et al. (2002) are, within errors, similar to those reported
by the authors. Together with the EXAFS fits of the NaBr aqueous solution, they confirm that Br
speciation in aqueous solution at room conditions is dominated by a six-fold coordinated
hydration shell with the H-O bond of the water molecule radially aligned towards the Br ion
(Ferlat et al., 2001; Ramos et al., 2000).
EXAFS spectra collected on NS2 and haplogranite glasses at room conditions display
distinct oscillations, with a new feature at 2.2 $Å^{-1}$ in both glass samples and amplitudes nearly
out of phase after 2 $Å^{-1}$ compared to the NaBr and KBr aqueous solutions (Fig. 5). Different
combinations of Br-Na and Br-O scattering paths were tested to constrain the local structural
environment of Br in the silicate glasses. Models considering individually either the Br-Na or Br-
O paths do not provide a reasonable fit of the EXAFS oscillations and the simultaneous
contribution of Br-Na and Br-O bond is required to reproduce the experimental spectra. The
EXAFS-derived parameters suggest that Br in NS2 and haplogranite glasses is coordinated to an
average of 6 Na cations in the first shell at an average distance of ~2.95 Å and 6 O second
neighbors located at ~3.4 Å (Table 3).  The fitted Br-Na bond length is consistent, within errors,
with theoretical Br-Na distances in crystalline NaBr (2.987 Å, Deshpande, 1961) and is close to
that fitted for aluminosilicate glasses in a previous study (Cochain et al., 2015), suggesting Br is
incorporated in the silicate glasses in a "salt-like" structure, similar to NaBr.  The similarities





between the structural parameters fitted for anhydrous NS2 and hydrous haplogranite (3.3 wt%
$H_2O$) glasses also suggest that the nearest environment of Br remains largely anhydrous in
glasses containing relatively low water contents and that the O second neighbors may be from
the silicate network rather than distinct $H_2O$ or OH groups. Attempts to include the effect of
Br···H-O bonds in the fitting model by taking into account multiple scattering Br···H-O paths
instead of Br-O correlations only resulted in a systematic decrease of the fit quality (higher *R*-
factor). The sole difference between the two glasses is the presence of a pre-edge feature at
~13.468 keV in the haplogranite glass (Fig. 4). Such features have been attributed to the 1s to 4p
electronic transitions in Br (Burattini et al., 1991) and reported in several covalently bonded
and/or reduced Br-bearing compounds, including HBr, $Br_2$, and $CHBr_3$ (D'Angelo et al., 1993;
Feiters et al., 2005). While Evans et al. (2007) suggested that this feature could arise from partial
Br reduction in the presence of remaining carbon material in the sample from the synthesis,
changes in the local site symmetry around Br could also contribute to the development of such
feature. Recent HERFD-XAS measurements conducted on silicate glasses however demonstrate
that this feature is absent in basaltic and andesitic glasses and hence, specific to the structure of
granitic glass compositions (Louvel et al., 2020).

**3.2.2 High P-T aqueous fluids and hydrous silicate melts**

Bromine K-edge XANES spectra of high *P-T* aqueous fluids (3 wt% NaBr solution,

fluids at equilibrium with haplogranite melt and water-dominated fluids containing < 50 wt%
dissolved NS2) all share a shape very similar to that of the NaBr aqueous solution at room
conditions, suggesting a similar local structure of Br in $H_2O$-dominated phases at elevated T-P
(Fig. 4 and Fig. 5). Differences in the shape of the XANES spectra become more pronounced for



the hydrosilicate liquids with >50 wt% dissolved Si and Na and the hydrous NS2 melt (Fig. S2).
Although the maximum of the white line remains at 13.478 keV, it broadens and decreases in
amplitude compared to the 3 wt% NaBr aqueous solution. Also, the first post-edge resonance is
shifted toward either higher (13.504 keV) or lower (13.487 keV) energies compared to the
aqueous fluids. We believe these changes may be indicative of the progressive incorporation of
Na in the local structure around Br. These modifications of Br coordination environment are also
noticeable in the EXAFS oscillations (Fig. 5): while Br-bearing aqueous fluids mostly show a
decrease of the amplitude of the oscillations with increasing $P$-$T$, they are shifted to higher
distances (*i.e.* from 2.6 to 2.8 Å$^{-1}$ for the first oscillation) for the 60 wt% NS2 fluid. Moreover,
the NS2 melt bears closer resemblance to the NS2 and haplogranite glasses, sharing similar
oscillations at 2.2 and 3.2 Å$^{-1}$.

The structural parameters derived from the quantitative EXAFS analysis are reported in

Table 4. Comparably to room conditions, the EXAFS spectra of 3 wt% NaBr aqueous solution at
high pressure-temperature conditions are well reproduced by an octahedral hydration shell
including multiple-scattering contributions from the Br$\cdots$H-O cluster (Fig. 5). Br-O coordination
numbers and distances are 6.4 ± 1.1 and 3.40 ± 0.07 Å at 450 °C and 0.6 GPa, indicating the
persistence of the 6-fold coordinated hydration shell up to high temperatures. This observation
contrasts with results from a number of classical EXAFS studies performed at lower pressures (<
650 bar at 450 ºC) that reported significant reduction in the number of water molecules around
Br at supercritical conditions (Wallen et al., 1997; Da Silva et al., 2009). These differences
reflect the role of pressure (or fluid density) in stabilizing the hydration shell around Br by
increasing the dielectric constant of the solvent with increasing $P$ and density (Pan et al., 2013;
Sverjensky et al., 2014), as also predicted for other ions such as Li$^+$ (Jahn and Wunder, 2009)





and $Ti^{4+}$ (van Sijl et al., 2010) in molecular dynamics studies. An exception to this trend are the
experimental results of Mayanovic et al. (2001), who reported a decrease by >60% of the number
of water molecules in the solvation shell of both Br aqua ions and $ZnBr_4^{2-}$ complexes in 1 m
$ZnBr_2$ - 6 m NaBr aqueous solution from ambient conditions to 500 °C and 0.5 GPa. The reason
for this discrepancy is unclear at this point and additional studies on the speciation of Br in
aqueous electrolytes will be necessary to explain the disagreement.

There are no significant changes in Br speciation in the aqueous fluids equilibrated with

haplogranitic melts, which contain only few wt% of dissolved silicate components, and in fluids
containing up to 30 wt% dissolved NS2 (Fig. 6; Table 4). The first noticeable changes are found
for fluids containing 50 wt% dissolved NS2, with a small decrease of the average Br
coordination number ($N_{Br…H-O}$) to ~ 4.7 compared to more dilute fluids (~6.0). While this value
stays within errors from the other compositions, the reduction of the hydration shell might define
the onset of Br-Na complexation with increasing amount of Na dissolved in the fluid. This
hypothesis was tested by introducing a Br-Na contribution in the fitting model for the high
temperature data, but this resulted in a decrease of the overall fit quality. The formation of Br-Na
complexes and the partial dehydration of Br however becomes evident with further increase of
the solute content to 60 wt% dissolved NS2 in the fluid (Table 4). For this composition, the best-
fit model is consistent with the presence of ~ 3 Na atoms and 4 to 5 $H_2O$ molecules (or OH
groups) in the nearest environment of Br, at 480 °C and 1.5 GPa and 610 °C and 2.2 GPa. In the
NS2 hydrous melt (10 wt% $H_2O$), the number of Na neighbors further increases to ~ 6 whereas
the number of oxygens remains similar to that of the 60 wt% NS2 fluid (~ 3.4). This increase in
the number of Na neighbors compared to the 60 wt% NS2-bearing fluid suggests that the nearest
environment of Br progressively approaches the local structure observed in the NS2 glass. Yet,





the Br local environemnt remains hydrated, in contrast to the NS2 and haplogranite glasses.
Based on results from FTIR and $^{29}$Si NMR studies showing that molecular $H_2O$ is favored in
aluminosilicate and sodium silicate glasses as the amount of dissolved water increases (Stolper,
1982; Uchino et al., 1992; Xue and Kanzaki, 2004; Behrens and Yamashita, 2008), we suggest
that molecular $H_2O$, rather than OH groups, would be present around Br in the hydrous NS2
melt. Moreover, we cannot exclude that distinct "fluid-like" $Br(H_2O)_6$ and "glass-like" $BrNa_6$
complexes coexist in the hydrous melt as $[yBr(H_2O)_6 + xBrNa_6]$ moieties, as the average signal
of these structures could not be distinguished from $[BrNa_y(H_2O)_x]$ clusters by XANES or
EXAFS.

**4. Implications for the transport and recycling of halogens in subduction zones**

The new partitioning and speciation data derived for bromine in the present study provide

direct insights on the recycling and transport mechanisms of halogens (Cl, Br and I) in
subduction zones. Our results suggest that the mobilization of Br (and likely Cl, Br and I) in
subduction zones is affected by the chemistry of the slab-derived mobile phases. These phases, in
turn, are essentially controlled by the slab composition and the depth of fluid extraction and
hence, by the pressure and temperature conditions (Schmidt and Poli, 1998; Manning, 2004;
Schmidt et al., 2004; Hermann et al., 2006; Bebout, 2007; Keppler, 2017). Figure 6 illustrates a
gradual transition of Br speciation from hydrated species $[Br(H_2O)_6]^-$ to $[BrNa_x(H_2O)_y]$ clusters
with various stoichiometries (or mixture of $[Br(H_2O)_6]$ and $BrNa_6$ moieties) as the fluid
composition evolves from diluted aqueous fluids such as those released by continuous
metamorphic dehydration of the slab (< 15 wt% dissolved solutes, Manning, 2004; Rustioni et
al., 2019) to Si/Na-rich supercritical fluids that form owing to enhanced solubility of silicate





minerals at depth and/or granitic melts produced by fluid-assisted melting of subducted
sediments (Hermann et al., 2006; Skora and Blundy, 2010). The increasing similarities in the
local structure of Br in aqueous fluids containing large amounts of dissolved alkali-silica (> 12.5
wt% Na) and the hydrous melts (Fig. 6) is consistent with the progressive decrease in the Br
fluid-melt partition coefficients ($D_{Br}^{f/m}$) as with *P-T* increase as observed in this study (Fig. 3)
and by Bureau et al. (2010). Sodium complexation with Br is thus an efficient mechanism that
enables not only aqueous fluids but also hydrosilicate liquids and hydrous melts to carry
significant amounts of Br at depth.

General similarities between Cl, Br and I speciation in aqueous solutions and silicate

glasses (Evans et al., 2008; McKeown et al., 2011,2015; Shermann et al., 2010) suggest that the
speciation and partitioning trends found in our study for Br may extend to Cl and I. Therefore,
while pore fluids and early dehydration fluids should release large amounts of halogens to the
fore-arc and the mantle wedge (100 – 200 km depth), hydrous slab melts and supercritical fluids
play a critical role in recycling the residual halogens dragged by the subducting slabs to greater
depths. Such efficient recycling, where most of the Cl and Br subducted is transfered to the
mantle wedge and ultimately returned to the surface through arc magmatism, is further supported
by recent quantification of halogens in subducted sediments, serpentinites and altered oceanic
crust. Mass balance calculations indeed show a close match between worldwide yearly influx to
the mantle wedge, $\sim$ 13-15 x10$^3$ kt/yr Cl and 5-70 kt/yr Br, and calculated outflux as HCl and
HBr at volcanic arcs, $\sim$ 3-22 x10$^3$ kt/yr Cl and 5-15 kt/yr Br. (Barnes et al., 2018; Chavrit et al.,
2016; Kendrick et al., 2013; Pyle and Mather, 2009). In comparison, iodine degassing at volcanic
arcs is less well constrained, making it more difficult to assess its fate in the subduction factory
(e.g., Bureau et al., 2016). The small imbalances remaining between Cl and Br input and output



fluxes may arise from difficulties in quantifying halogens loss to the fore-arc and crustal
hydrothermal systems. Recent reports of halogens enrichment in oceanic islands basalts also
point out to the subduction of a noticeable fraction of F, Cl, Br and I to greater depth, to an
extent that is still to be quantified (Barnes et al., 2018; Hanyu et al., 2019; Kendrick et al., 2017).

**5. Conclusions**
*In-situ* SXRF and XAS have been applied to quantify Br fluid-melt partition coefficients
and speciation in aqueous fluids, hydrosilicate liquids and hydrous melts up to 840 °C and 2.2
GPa.Above all, our experimental results demonstrate how changes in speciation, from hydrated
ions in aqueous fluids to 'salt-like' structures in hydrous melts, may facilitate the uptake of high
amounts of Cl, Br and probably I by subduction zone fluids, regardless of their composition.
Significant efforts are however still needed to accurately quantify halogen's cycling from the
surface to the deep Earth and back. Especially, new experiments investigating the solubility of
halogens in subduction zone fluids and the capacity of high-pressure minerals (e.g., micas, Ti-
clinohumite, apatite, nominally anhydrous minerals but also carbonates) to incorporate these
elements are still necessary to evaluated the amounts that may be returned to the volcanic arc or
retained in the slab.


**Acknowledgements:** We would like to thank M. Doebeli and J. H. Seo for conducting the RBS
and LA-ICPMS analysis, respectively. The Paul Scherrer Institute (PSI) and the Swiss Light
Source (SLS) are acknowledged for providing beamtime for the experiments. This work was
supported by the Swiss National Science Foundation (grants 200021-120575 and 200020-132208



to CSV) and by the Swiss Academy of Sciences (SATW) and the Ministères des Affaires
étrangères et européennes (MAEE) et de l'Enseignement Supérieur et de la Recherche (MESR)
through the Partenariat Hubert Curien (PHC).

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





**List of Tables and Figure captions:**

**Table1.** Synthesis conditions and chemical compositions of the $Na_2Si_2O_5$ (NS2) and haplogranite (Hpg)
glasses employed as starting materials in this study.

**Table 2:** Bromine fluid-melt partition coefficients at different P-T conditions. Calculated fluid and melt
compositions and densities are also reported.

**Table 3.** Structural parameters derived from Br K-edge EXAFS analysis for the reference aqueous
solutions and silicate glasses at ambient conditions.

**Table 4:** Br K-edge EXAFS analysis of experimental high P-T fluids with various compositions.

**Figure 1.** Microphotographs of the compression chamber of the HDAC showing the Haplogranite - $H_2O$
(A, B, C) and NS2 - $H_2O$ (D, E, F) systems at the indicated pressure and temperature conditions. Images
are taken through the diamond along the X-ray path. A) Haplogranite glass and 3 wt% NaBr aqueous
solution at room conditions; B) globulus of hydrous silicate melt in equilibrium with the aqueous fluid; C)
supercritical liquid (single fluid phase); D) NS2 glass and 3 wt% NaBr aqueous solution at room
conditions; E) supercritical liquid (low temperature supercriticality); F) hydrous NS2 melt coexisting with
aqueous fluid (high temperature subcriticallity).

**Figure 2.** 2D-SXRF Br $K_\alpha$ intensity maps of Run 1 showing the distribution of Br between coexisting
aqueous fluid and haplogranite melt at different P-T conditions. The fluid:glass ratio refers to the wt
fraction calculated from the volumetric proportions of loaded glass and sample chamber. The white
dashed line delimits the edge of the Re gasket.





Br-enriched phases appear in red and yellow, Br-depleted areas in blue and green. At the beginning of the
experiment (A), all the Br is concentrated in the glass. After the glass melts (B), Br strongly partitions
into the fluid phase ( $D_{Br}^{f/m}$ = 8.07 ±0.79). As temperature increases, the Br concentration in the melt
increases while the Br concentrations in the fluid decreases (C). At 821°C - 0.9GPa, the $I_f/I_m$ ratio appears
homogeneous as the Br concentrations per volume are almost similar ($I_f/I_m$ = 1.3 ±0.1). However, per
weight, Br still partitions preferentially into the fluid ( $D_{Br}^{f/m}$ = 2.02 ±0.14).

**Figure 3.** Evolution of the Br partition coefficients $D_{Br}^{f/m}$ with increasing temperature at different
pressure conditions. The different symbols and colors account for separate experimental runs involving
different glass proportions. The errors reported on $D_{Br}^{f/m}$ take into account the uncertainties on pressure
determination (10%) and on the determination of fluid and melt composition and density from previous
studies. The partition coefficients from Bureau et al. (2010) and Cadoux et al. (2018) are shown for
comparison. For Cadoux et al., note that both average values from several experiments ( $D_{Br}^{f/m}$ = 20.2
±1.2) and minimum value for a single experiment ( $D_{Br}^{f/m}$ = 8.6) are reported.

**Figure 4.** Normalized Br K-edge XANES spectra collected on Br-bearing silicate glasses, aqueous fluids
and hydrous silicate melts at various pressure and temperature conditions. Spectra are offset for clarity.
The vertical dashed line is a visual guide to appreciate phase shifts. The black arrow shows the pre-edge
feature in the haplogranite glass spectrum corresponding to the 1s to 4p transition in Br (Burattini et al.,

788    1991).


**Figure 5.** Normalized $k^1$-weighted EXAFS oscillations of the investigated Br-bearing samples (black
solid lines) and corresponding least-square fits (blue dashed lines). Spectra are off-set vertically for
clarity. The pressure and temperature conditions and the compositions are reported right to each spectrum.






**Figure 6.** Evolution of bromine coordination numbers with oxygen (from $H_2O$ molecules) and sodium
($N_{Br\cdots H\text{-}O}$ and $N_{Br\text{-}Na}$) as a function of fluid composition (*i.e*, the weight fraction of NS2 dissolved in the
fluid) along the NaBr aqueous solution – NS2 join. The gray field shows the detection limit (DL) for Br-
Na complexes, which corresponds to the maximum Br-Na coordination number ($N_{Br\text{-}Na}$) determined for
3wt% NaBr aqueous solution at ambient conditions and 450 °C (DL < 1.5 atoms).





























**Table 1.**

| Sample | Synthesis conditions | | Br[1] (wt%) | $Na_2O$[2] (wt%) | $SiO_2$[2] (wt%) | $Al_2O_3$[2] (wt%) | $K_2O$[2] (wt%) | $H_2O$[3] (wt%) | ASI[4] | Analytical method |
|---|---|---|---|---|---|---|---|---|---|---|
| | T (°C) | P(GPa) | | | | | | | | |
| NS2-Br1 | 1200 | 0.5 | 4.01 4.10 | 32.0 | 63.9 | | | | | EMPA LA-ICPMS |
| Hpg-Br2 | 1200 | 1.5 | - 0.96 | 7.1 | 74.1 | 9.2 | 3.7 | 3.3 | 0.57 | EMPA RBS |
| Hpg-Br3 | 1200 | 1.5 | 0.89 | 7.4 | 75.0 | 9.4 | 3.8 | 3.3 | 0.57 | EMPA |

**Notes: EMPA** = Electron Microprobe Analyses; **LA-ICPMS** = Laser-ablation Inductively Couple Plasma Mass Spectrometry;
**RBS** = Rutherford Backscattering Spectroscopy.
[1]Standard deviations (1σ) are 0.04 wt% for RBS analysis, 0.3 wt% for LA-ICPMS and 0.03 wt% for EMPA analysis.
[2]Average from 10 to 25 analyses performed on each glass composition. Standard deviations (1σ) are < 0.1 wt% for $Na_2O$, $Al_2O_3$ and
$K_2O$ and < 0.3 wt% for $SiO_2$.
[3]Nominal $H_2O$ concentration (not analyzed).
[4]Aluminum Saturation Index ASI = $\frac{Al_2O_3}{Na_2O+K_2O}$ (in moles).





**Table 2.**

| $X_g{}^1$ | T (°C) | P (GPa)[2] | $H_2O$ in melt (wt%)[3] | Melt density $\rho_m$[4] | Transmission in melt $T_m$ | Silicates in fluid (wt%)[5] | Fluid density $\rho_f$[7] | Transmission in fluid $T_f$ | $I_{Br}^f / I_{Br}^m$ | $D_{Br}^{f/m}$ |
|---|---|---|---|---|---|---|---|---|---|---|
| **Haplogranite – $H_2O$** | | | | | | | | | | |
| *Run 1* | | | | | | | | | | |
| | 592 | 0.7 | 7.1 ±0.8 | 2.24 | 0.66 | 2.5 ±0.5 | 0.94 | 0.96 | 4.9 | **8.1 ±0.8** |
| *0.76* | 694 | 0.8 | 7.7 ±0.9 | 2.23 | 0.67 | 5.3 ±1.2 | 0.97 | 0.95 | 2.6 | **4.2 ±0.2** |
| | 821 | 0.9 | 8.0 ±1.0 | 2.23 | 0.67 | 10.3 ±2.3 | 0.99 | 0.95 | 1.3 | **2.0 ±0.1** |
| *Run 2* | | | | | | | | | | |
| | 645 | 0.9 | 9.1 ±1.1 | 2.22 | 0.67 | 5.5 ±1.1 | 1.02 | 0.95 | 10.0 | **15.3 ±1.0** |
| *0.82* | 710 | 1.1 | 11.1 ±1.4 | 2.20 | 0.68 | 11.1 ±2.4 | 1.09 | 0.94 | 5.4 | **7.9 ±0.5** |
| | 840 | 0.9 | 7.9 ±1.0 | 2.23 | 0.67 | 10.8 ±2.4 | 0.98 | 0.95 | 2.8 | **4.4 ±0.3** |
| *Run 3* | | | | | | | | | | |
| | 610 | 1.2 | 13.3 ±1.7 | 2.18 | 0.69 | 7.7 ±1.6 | 1.13 | 0.95 | 4.6 | **6.4 ±0.3** |
| *0.72* | 730 | 0.65 | 6.0 ±0.7 | 2.25 | 0.66 | 3.9 ±0.9 | 0.88 | 0.96 | 2.3 | **4.1 ±0.4** |
| | 800 | 0.2 | 2.4 ±0.2 | 2.26 | 0.64 | 0.7 <0.1 | 0.49 | 0.98 | 1.6 | **4.8 ±0.3** |
| **Haplogranite – 3 wt% NaBr aqueous solution** | | | | | | | | | | |
| *Run 4* | | | | | | | | | | |
| *0.70* | 740 | 1.7 | 19.5 ±2.8 | 2.11 | 0.72 | 12.8 ±0.8[6] | 1.20 | 0.94 | 7.2 | **9.7 ±0.6** |
| *Error (unless indicated)* | | ±0.1 | | ±0.01 | ±0.07 | | ±0.04 | ±0.01 | ±0.1 | |

**Notes:**
[1] Initial weight fraction of glass in the loading.
[2] Maximum uncertainty on pressure were of 10%.
[3] $H_2O$ solubility in the haplogranite melt calculated from the solubility data of Mysen and Wheeler (2000).
[4] Melt density (in $g.cm^{-3}$) calculated as a function of P-T conditions and melt composition using Malfait et al. (2014).
[5] Solubility of silicate components ($SiO_2$, $Na_2O$, $Al_2O_3$ and $K_2O$)in the aqueous fluid coexisting with haplogranite
melt calculated from the albite solubility data of Anderson and Burnham (1983).
[6] Silicate solubility in the aqueous fluid estimated from Wohlers et al. (2011) for P > 1.2 GPa.
[7] Fluid density (in $g.cm^{-3}$) calculated as a function of P-T conditions from the data of Mantegazzi et al. (2013)








**Table 3.**

| Composition | Oxygen (O) | | | Sodium (Na) | | | |
|---|---|---|---|---|---|---|---|
| *Aqueous solutions*[1] | | | | | | | |
| | $N_{Br\cdots H\text{-}O}$ | $R_{Br\cdots H\text{-}O}$ (Å) | $\sigma^2$ (Å$^2$) | | | | *R-factor* |
| **3 wt% NaBr-H$_2$O** | 5.9 ±0.7 | 3.37 ±0.04 | 0.02 | | | | 0.04 |
| **2.3 wt% KBr-H$_2$O.[3]** | 5.7 ±0.8 | 3.30 ±0.03 | 0.02 | | | | 0.06 |
| *Silicate glasses*[2] | | | | | | | |
| | $N_{Br\text{-}O}$ | $R_{Br\text{-}O}$ (Å) | $\sigma^2$ (Å$^2$) | $N_{Br\text{-}Na}$ | $R_{Br\text{-}Na}$ (Å) | $\sigma^2$ (Å$^2$) | *R-factor* |
| **NS2 glass** | 5.2 ±2.4 | 3.45 ±0.09 | 0.02 | 5.3 ±1.8 | 2.99 ±0.09 | 0.03 | 0.25 |
| **Haplogranite glass** | 6.1 ±3.6 | 3.39 ±0.03 | 0.02 | 5.9 ±1.8 | 2.94 ±0.03 | 0.03 | 0.21 |

**Notes:** N = Br coordination number ($N_{Br\text{-}O}$ or $N_{Br\text{-}Na}$); R = Br-neighbor (Na or O) mean distance (Å); $\sigma^2$ = squared Debye-Waller
factor (Å$^2$); *R-factor* = goodness of the fit; $S_0^2 = 1$;
[1] Hydration shell (Br··H-O)
[2] Br coordinated to oxygens from the silicate network (next-nearest coordination shell).
[3] Ferlat et al. (2002), 0.2m KBr-H$_2$O for comparison.























**Table 4.**

| Composition | T (°C) | P (GPa) | $N_{Br \cdots H-O}$ | $R_{Br \cdots H-O}$ (Å) | $\sigma^2$ (Å²) | $N_{Br-Na}$ | $R_{Br-Na}$ (Å) | $\sigma^2$ (Å²) | R-factor |
|---|---|---|---|---|---|---|---|---|---|
| *3 wt% NaBr aqueous solution* | | | | | | | | | |
| | 25 | 0 | 5.9 ± 0.7 | 3.37 ±0.04 | 0.02 | bdl[1] | | | 0.04 |
| | 320 | 0.2 | 6.3 ± 1.8 | 3.36 ±0.05 | 0.04 | bdl | | | 0.17 |
| | 450 | 0.6 | 6.4 ±1.1 | 3.40 ±0.07 | 0.05 | bdl | | | 0.19 |
| *Br-bearing aqueous fluids* | | | | | | | | | |
| 1.2 wt% Hpg[3] | 475 | 1 | 5.4 ±0.9 | 3.33 ±0.03 | 0.05 | bdl | | | 0.13 |
| 5 wt% Hpg | 680 | 0.8 | 5.7 ±1.1 | 3.30 ±0.04 | 0.06 | bdl | | | 0.12 |
| 0.6 wt% Hpg | 750 | 0.2 | 5.0 ±1.6 | 3.33 ±0.06 | 0.06 | bdl | | | 0.30 |
| 30 wt% NS2[2] | 190 | n.d. | 6.7 ±1.4 | 3.38 ±0.03 | 0.04 | bdl | | | 0.14 |
| | 320 | n.d. | 5.7 ±1.4 | 3.37 ±0.09 | 0.04 | bdl | | | 0.22 |
| 50 wt% NS2 | 580 | 1.1 | 4.7 ±1.5 | 3.35 ±0.15 | 0.04 | bdl | | | 0.25 |
| *Br-bearing melt-like fluids* | | | | | | | | | |
| 60 wt% NS2 | 480 | 1.5 | 3.6 ±1.5 | 3.47 ±0.05 | 0.01 | 2.5 ±1.2 | 3.10 ±0.06 | 0.01 | 0.23 |
| | 610 | 2.2 | 4.8 ±2.4 | 3.45 ±0.05 | 0.03 | 2.6 ±0.9 | 3.06 ±0.06 | 0.03 | 0.20 |
| NS2 melt (10 ±1 wt% $H_2O$) | 710 | 0.4 | 3.4 ±1.6 | 3.36 ±0.03 | 0.02 | 6.6 ±2.1 | 2.91 ±0.03 | 0.05 | 0.24 |

**Notes**: N = Br coordination number (dissociate as $N_{Br \cdots H-O}$ and $N_{Br-Na}$); R = Br-neighbor mean distance (Å); $\sigma^2$ = squared Debye-
Waller factor (Å²); *R-factor* = goodness of the fit; $S_0^2$ = 1.
[1]bdl = below detection limit. Detection limit corresponds to the maximum Br-Na coordination number determined for 3 wt%
NaBr aqueous solution at ambient conditions.
[2]wt% NS2 indicates the amount of dissolved NS2 in the single phase fluid calculated from the mass of $H_2O$ and NS2 glass.
[3]wt% Hpg refers to the amount of dissolved silicate in the fluid coexisting with haplogranite melt calculated as in Table 3.
Errors in temperature and pressure are ±2 ºC and 10%, respectively. Errors in the composition of the analyzed fluids are within
5% (Table 2).



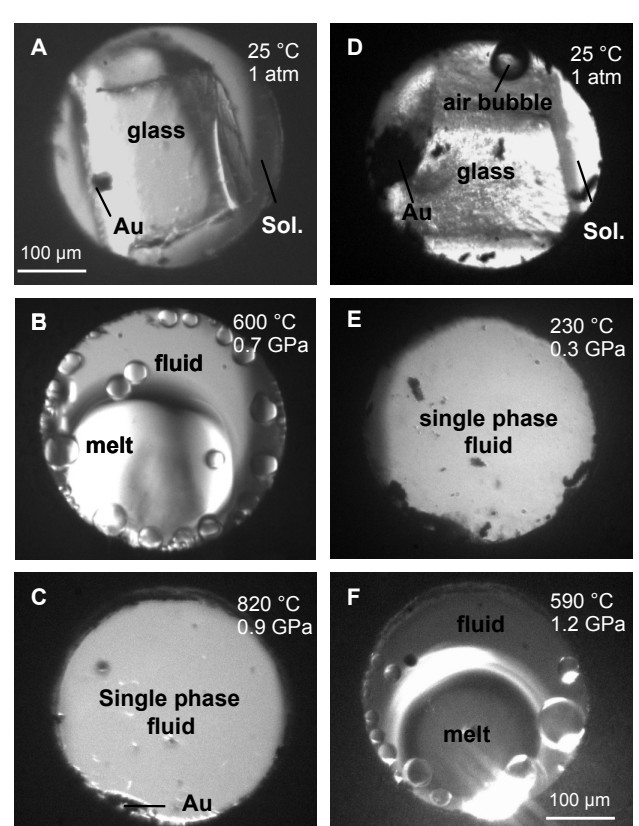


**Figure 1**






**Run 1 – Fluid:Glass ratio = 0.76**

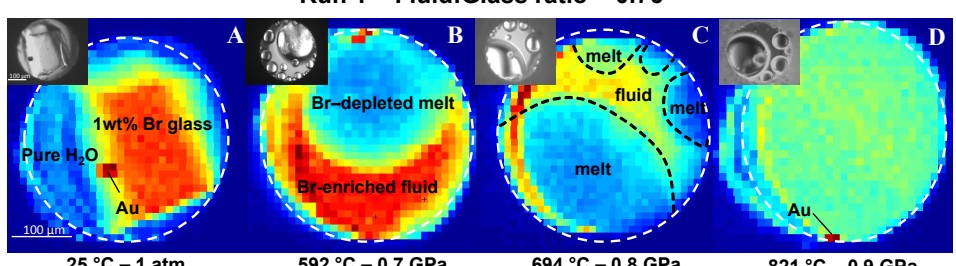



**Figure 2**











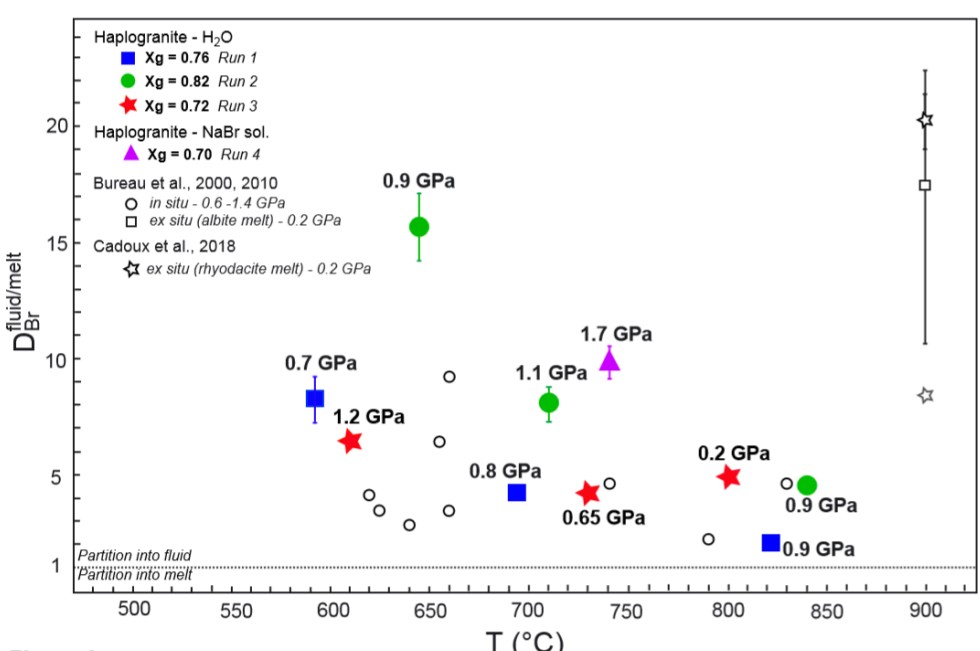

**Figure 3**





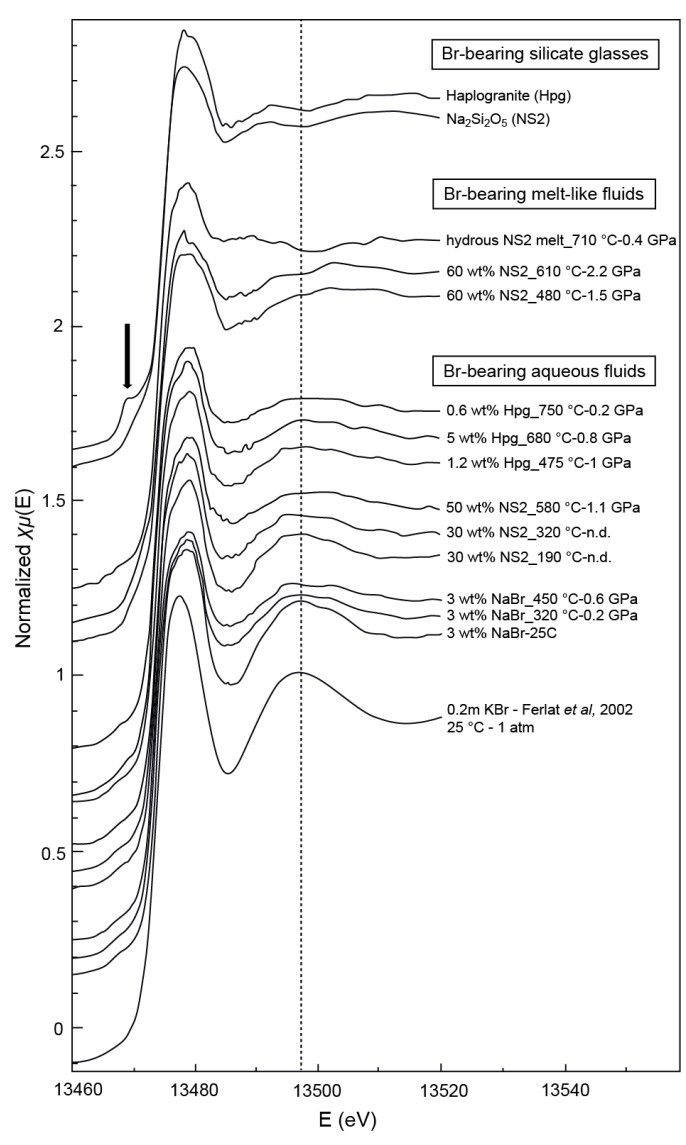



**Figure 4**




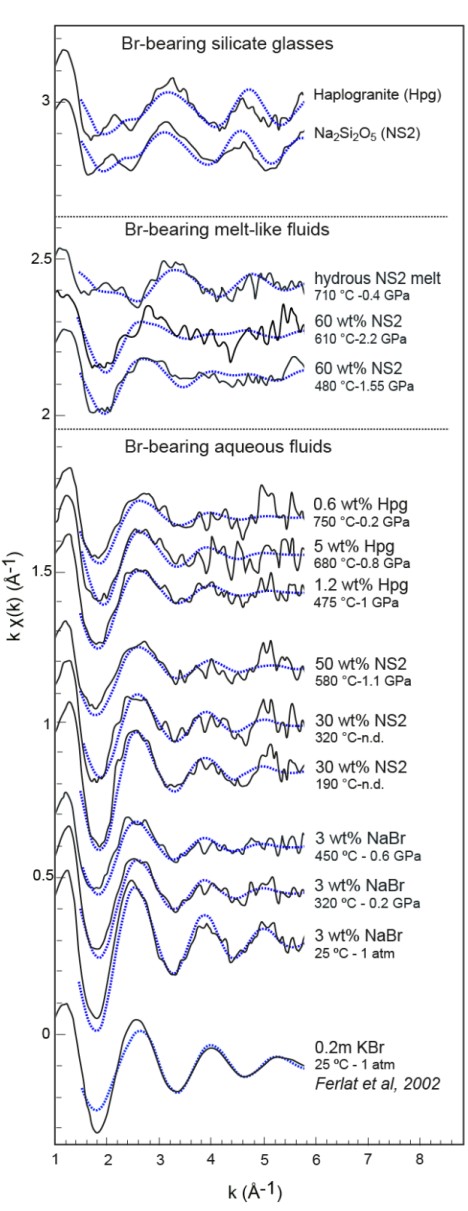



**Figure 5**






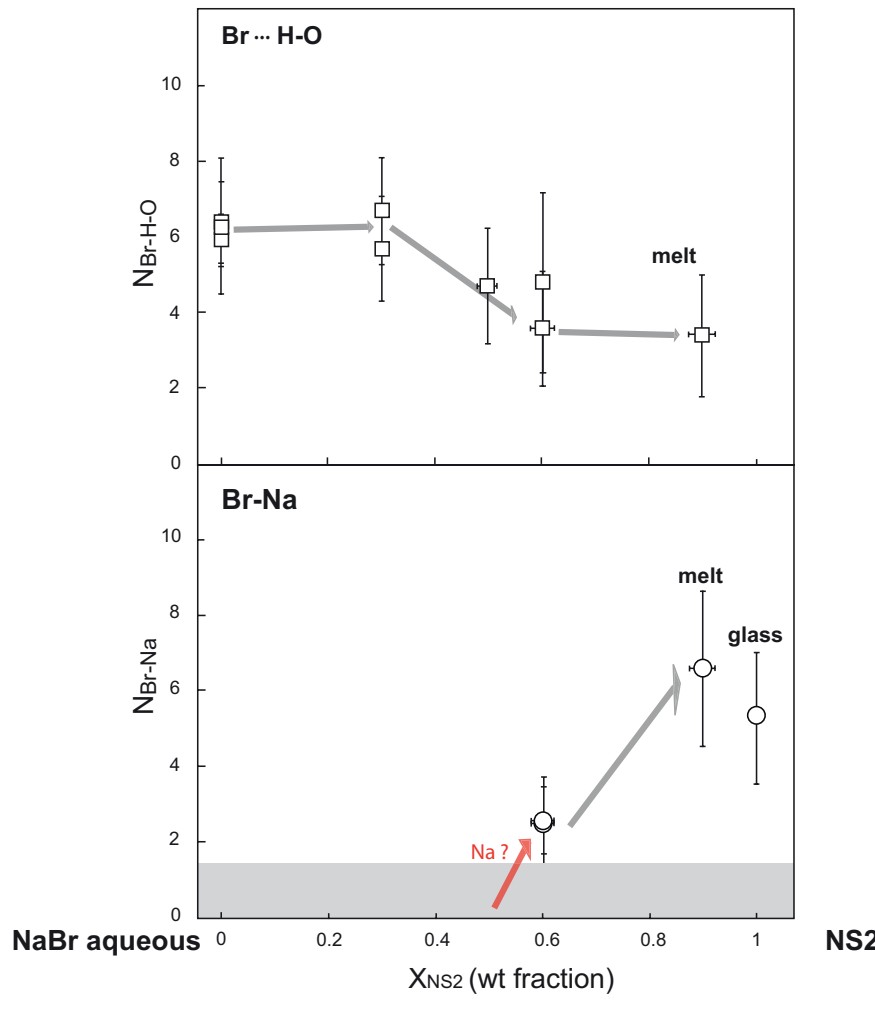



**Figure 6**