# Peer review of "Bromine speciation and partitioning in slab-derived aqueous fluids and silicate melts and implications for halogen recycling in subduction zones"

_Solid Earth, 2019_

## Referee Comment (RC1) · Anonymous Referee #1 · 18 Feb 2020

General comments This manuscript presents two experimental studies by combining diamond anvil cells experiments with (1) in situ synchrotron X-Ray fluorescence analysis and (2) in situ XANES and EXAFS spectroscopies. The general aim of the study is to investigate the partitioning of Bromine between silicate melts and aqueous fluids and the solution mechanisms of bromine in such fluids. The subject of the study is relevant because it reports very important questions about the recycling of halogen elements in subduction zones (bromine and chlorine because bromine is considered to be a good analogue for chlorine). For that reason I believe that this study should be published but not in its present form, for the following reasons: The results about partitioning are equivalent to those of a previous published study by another group (nothing new) and

the discussion about one low pressure result (at 2 GPA) is not convincing (see specific comments). The results about bromine speciation in high pressure fluids are new and they deserve to be published, unfortunately the partitioning of Br is not measured for the same chemical system (haplogranite, HPG) than its speciation ($Na_2Si_2O_5$, NS2), which makes any comparison difficult. Therefore I would recommend to delate the part about partitioning, or at least to provide convincing explanation (see specific comments), and to focus on the speciation results. Specific comments Br partition coefficients (D) are measured in situ for HPG system within the range 0.2 – 1.7 GPa and 592 – 840°C: they are ranging from 4.1 to 15.3, they fall in the same range than those from Bureau et al., 2010, for similar conditions (0.66 –1.7 GPa, 590-890°C, D from 2.18 to 9.2). However, if one plots all results in a diagram D versus pressure, data exhibits a lot of dispersion and no real relationship, as it should be expected (i.e. an increase of D with decompression due to degassing). This is not discussed at all. About the same value of D is obtained at 0.9, 0.8, 0.65, 0.2 GPa, respectively 4.4, 4.2, 4.1, and 4.8. High values of D are obtained at high investigated pressure (15 at 0.9 GPa and 9.7 at 1.7 GPa) where unity would be expected due to imminent total miscibility. Why? Such a discrepancy may be due to the pressure determination. The authors use the diffraction of gold and its equation of state. However, it is well known that gold diffraction is not a good tool to for low pressure determination, as an example see Heinz and Jeanloz, JAP, 1983, where the first measurements are performed at 4.42GPa. Furthermore, in a recent intercomparison of the use of EOS for pressure determination (including Au), Ye et al., JGR, 2017, it is conclude that at high temperature, accuracy cannot be better than 1 GPa, from 2.5 GPa up to 140. For that reason, the discussion from line 278 to 294 is not relevant and should be suppressed.

---

## Referee Comment (RC2) · Anonymous Referee #2 · 22 Feb 2020

This paper sets out to explore the partitioning of bromine (and therefore Chlorine by proxy) between silicate melts and aqueous fluids and to also determine the speciation of bromine within these fluids. This is an important area to investigate, as pointed out by the authors, to understand how subduction zones constrain the recycling of trace elements and metals, and to quantify the halogen fluxes to the atmosphere via volcanic degassing, which is vital for understanding climatic influences they may subsequently cause. Understanding the effects of halogens and other volatiles on these processes is vital. Therefore, this study should be published as it adds partitioning and speciation data to the literature; however, some minor concerns should be clarified.

[Figure]

Firstly, the authors set about trying to attain Br partitioning between a fluid phase (water or NaBr solutions) and a haplogranite melt by performing diamond anvil cell experiments. I would like a little clarification on the following:

1) The results outlined in table 2 show that some silicates have been extracted from the silicate melt and dissolved into aqueous fluid, varying from 0.7% to 11.1% in H2O and 12.8% in the NaBr solution. This change in the composition may greatly influence the solubility and therefore partitioning of Br. As stated in this paper, Br complexation with alkalis is an efficient transfer mechanism, therefore, with up to 12.8% silicate components going into the fluid, including Na2O and K2O may drastically change the partitioning coefficients attained in this study. Can any limitations on this effect be noted in this study?

2) In Table 2, point 5, it is stated that solubility of silicate components into the aqueous fluid is calculated from albite solubility data of Anderson and Burnham (1983) – how reliable is this? How similar in compositions are yours from theirs?

3) Please could you display the Br concentrations in both phases as well as the D values calculated?

4) Table 2- References such as Wohlers et al., 2012 are missing from the reference list

5) Partitioning was determined via the intensities of Br in the fluid and the melt (equation 1), with Figure 2 showing Br Ka intensity maps which are utilised to calculate Br concentrations. In this figure, the Au displays that its intensity is similar, if not greater than that of Br rich phases this could be a function of the gating used during analysis, as Br $K\alpha$ is 11.9, and Au Lß is 11.5 which might explain its relative intensity on par with the measured Br. This begs the question: what gating size was utilised in this analysis, and did it have any effect on the overall Br concentrations recorded?

6) Br concentration was determined in the starting composition mixtures by various techniques. Noting the limitations of measuring Br via EPMA due to the overlap with

Al (lines 104-106). However, by calculating the overlap of Br using the Lb line one can accurately measure Br efficiently; this can be used to check your RBS data and to be consistent between all measurements.

7) Partitioning of Br is not measured in the same composition as those studied for speciation, making any comparison challenging.

8) Once plotted all D values, versus P or T, shows little relationship and a lot of dispersion. Which makes it difficult in understanding how the partitioning of Br may vary along a subduction zone. Also, plotting D values in a density plot against P and T show no clear trends. Can this be discussed?

9) Can further discussion go towards how your results depict a change in Br partitioning with subduction?

10) Figure 3 – the label at the bottom of the graph is not correct – this shows that Br preferentially is partitioned into the fluid. Br should have partitioned into both phases, but preferentially into the fluid.

Unfortunately, I am not an expert in EXAF analyse and so cannot comment on the methodology employed here. However, whilst reading other papers such as Cochain et al 2015, they discuss the reference material that they employed in measuring Br speciation. Were any standards measured at the synchrotron under the conditions tested, or are all the results based on those referenced in the paper such as Ferlat et al 2002 etc? If so a plot or a discussion of how good the fit to each speciation might be useful.

Overall, I found that this paper is expanding our current knowledge of Br speciation by performing experiments between fluid and melt at higher pressures, and so should be published. If the above comments are addressed, I think that this will be a great addition to the literature in this area.

---

## Author Comment (AC1) · 7 Apr 2020

**Answer to Referee 1:**

First of all, we would like to thank Referee 1 for appreciating the relevance of our *in-situ* study to further the current understanding of halogen deep cycle.

The referee's concerns are addressed as follows:

**Comment 1:** The results about partitioning are equivalent to those of a previous published study by another group (nothing new) and the discussion about one low pressure result (at 0.2 GPa) is not convincing (see specific comments below). The results about bromine speciation in high pressure fluids are new and they deserve to be published, unfortunately the partitioning of Br is not measured for the same chemical system (haplogranite, HPG) than its speciation ($Na_2Si_2O_5$, NS2), which makes any comparison difficult. Therefore I would recommend to delete the part about partitioning, or at least to provide convincing explanation (see specific comments), and to focus on the speciation results.

**Answers:**
- While the in-situ Br partitioning experiments are not the first of their kind, they provide a unique opportunity for cross-checking experimental reproducibility and thus, we believe they deserve to be included in the manuscript. References to previous work by Bureau can be found throughout the manuscript and the favourable comparison between their and our studies further supports the reliability of the in-situ measurements. It should be stressed that in-situ measurements as those reported here and in rare previous work are extremely challenging, but the only reliable way to assess element partitioning and speciation at extreme P-T conditions, and thus any new data would be a valuable contribution to the field. **Therefore, we prefer to keep the partitioning experiments as part of the current manuscript.**
- Partitioning experiments involved haplogranite melts (Si, Al, Na, K), while Br speciation in melts could only be determined for sodium disilicate ($Na_2Si_2O_5$) due to insufficient Br concentrations in the haplogranite melt (400-2000 ppm Br). Yet, both XANES and EXAFS analyses (Figures 4 and 5; Table 3) show that Br local environment is very similar in the haplogranite and NS2 glasses. Thus, it can be expected that Br incorporation mechanism in both melts is similar and controlled by the presence of alkalis, either Na or K, and that all peralkaline silicate melts will have affinity to incorporate high amounts of Br under high P-T conditions. A note has been added in **Lines 479-482** to clarify this point and emphasize the similarities between the haplogranite and NS2 systems.
- We believe that underlining the differences between our low *P* $D_{Br}$ (4.8 at 800 °C, 0.2 GPa) and those of Bureau et al. and Cadoux et al (17.5-20.2 at 900 °C, 0.2 GPa) is of relevance to this study to **highlight that significant amounts of Br (and Cl) may be retained in degassed lavas, as reported in natural context.** The discussion has been modified to highlight this point **(Lines 356-364)**.

**Comment 2:** Br partition coefficients (D) are measured in situ for HPG system within the range 0.2 – 1.7 GPa and 592 – 840 °C: they are ranging from 4.1 to 15.3, they fall in the same range than those from Bureau et al., 2010, for similar conditions (0.66 –1.7 GPa, 590-890∘C, D from 2.18 to 9.2). However, if one plots all results in a diagram D versus pressure, data exhibits a lot of dispersion and no real relationship, as it should be expected (i.e. an increase of D with decompression due to degassing). This is not discussed at all.

**Answer:**
- Referee 1 was right to point out the dispersion of the data and the apparent lack of correlation with P. A similar lack of pressure dependency (as well as density, or composition) has previously been reported for vapour/brine partitioning of some metals (Cu, Au, Ag). It was suggested that large differences in the speciation of these elements in both phases could be

responsible for such a behaviour (Pokrovski et al., 2008). We thus suggest that the large differences in Br chemical and structural environment in between the coexisting fluid and melt phase could as well explain the scattered $D_{Br}$ in our study and the apparent lack of simple trends with P, density or dissolved silicate content, since the physical-chemical controls of Br in the two phases are very different due to the different speciation. We would also like to point out that with a single exception at 0.2 GPa, all our data were recorded at P > 0.5 GPa, where the existing studies have also shown that $D_{Br}$ values do not change significantly with pressure and exhibit a similar degree of scattering, between 1 and 10 (Bureau et al., 2010).

- We also note that additional discrepancies may arise from uncertainties on the estimation of the fluid composition, which will affect the calculated $D_{Br}$. We, for instance, recognize that the large $D_{Br}$ value obtained at 1.7 GPa is clearly off the trend, probably due to the fact that the fluid composition was calculated using the albite solubility data of Wohlers et al. (2011) instead of Anderson and Burnham (1983), to take into account the higher P conditions in this experiment.

- We have added an additional discussion in the revised manuscript, both in the Results section (**Lines 316-342**) and in the Methods section, where we provide additional details about uncertainties on fluid and melt compositions and how they translate to the $D_{Br}$ (**Lines 179-186 and 240-258**). We also agree with this referee that the discussion of the temperature effect in a single experimental run was of weak relevance in terms of partitioning behaviour, and hence we have removed it from the revised manuscript.

- The conclusions drawn from our partitioning experiments, however, remain unchanged: we confirm that although Br preferentially partitions into the aqueous fluid over silicate melt, high amounts of Br can yet be incorporated in hydrous granitic melts. To strengthen this argument, we added an estimation of Br concentration range in the high P-T melts of this study, calculated using the in-situ $D_{Br}$ and initial phase proportions (**Lines 339-340**).

**Comment 3:** About the same value of D is obtained at 0.9, 0.8, 0.65, 0.2 GPa, respectively 4.4, 4.2, 4.1, and 4.8. High values of D are obtained at high investigated pressure (15 at 0.9 GPa and 9.7 at 1.7 GPa) where unity would be expected due to imminent total miscibility. Why? Such a discrepancy may be due to the pressure determination. The authors use the diffraction of gold and its equation of state. However, it is well known that gold diffraction is not a good tool to for low pressure determination, as an example see Heinz and Jeanloz, JAP, 1983, where the first measurements are performed at 4.42GPa. Furthermore, in a recent intercomparison of the use of EOS for pressure determination (including Au), Ye et al., JGR, 2017, it is conclude that at high temperature, accuracy cannot be better than 1 GPa, from 2.5 GPa up to 140. For that reason, the discussion from line 278 to 294 is not relevant and should be suppressed.

**Answer:**
- The choice of gold as the in-situ pressure calibrant was motivated by its chemical stability in high P-T fluids and melts. Although we agree that the absolute accuracy of this method may not be better than 1 GPa (Ye et al., 2017), the actual relative precision is much better and is likely to be within 10% of the P value, as reported in Louvel et al. (2014). Moreover, the unit cell volume of Au displayed systematic changes as a function of increasing P-T in the HDAC, thus demonstrating that Au is sensitive to relatively small pressure changes during the run, and special care was taken in the HDAC alignment/centering to preserve the sample-detector distance to ensure the reliability of the unit cell volume variations. Therefore, we are confident that the relative pressure variations during the run are captured by the Au pressure calibrant. To further support the appropriate pressure determination, we emphasize that the phase relations, including miscibility, in the haplogranite-$H_2O$ system are within the P-T range reported for other alkali silicate systems (e.g. Paillat et al., 1992; Stalder et al., 2000).

- As mentioned above, additional discussion on the uncertainties of our calculations is now added both in the methods and result sections. Note that all fluid and melt properties (composition, density and effective transmission) were calculated assuming an uncertainty on pressure determination of 10% **(Lines 183-186 and 245-246).**

**Additional comments:**
- The title has been updated to specify the nature of fluids (ie., aqueous) and melts (ie., silicate).
- Supplementary materials have been incorporated into the main manuscript text to better provide the reader with as many details as possible and to better address the reviewers' concerns **(Lines 103-108; 115-118; 121-130).**
- A set of dashed lines was added to Figure 5 to underline the shift of EXAFS oscillations with change in composition. The caption of the figure was changed accordingly.

---

## Author Comment (AC2) · 7 Apr 2020

**Answer to Referee 2:**

First of all, we would like to thank Referee 2 for the relevant comments and for acknowledging the importance of the present study. The referee raised several questions about the partitioning experiments, to which we answer as follows:

**Comment 1:** The results outlined in table 2 show that some silicates have been extracted from the silicate melt and dissolved into aqueous fluid, varying from 0.7% to 11.1% in $H_2O$ and 12.8% in the NaBr solution. This change in the composition may greatly influence the solubility and therefore partitioning of Br. As stated in this paper, Br complexation with alkalis is an efficient transfer mechanism, therefore, with up to 12.8% silicate components going into the fluid, including $Na_2O$ and $K_2O$ may drastically change the partitioning coefficients attained in this study. Can any limitations on this effect be noted in this study?

**Answer:**
We did not observe Br-Na complexation until more than 50-60 wt% Si and Na were dissolved in the aqueous fluid (Table 3, Fig. 6), so we do not expect any complexation with alkalis to have occurred in the fluid in the partitioning experiments.
Further explanations for the scattered $D_{Br}$ values and the lack of pressure-density-composition trends are given in the answer to Referee 1 and have been added to the revised manuscript **(Lines 316-342).**

**Comment 2:** In Table 2, point 5, it is stated that solubility of silicate components into the aqueous fluid is calculated from albite solubility data of Anderson and Burnham (1983) – how reliable is this? How similar in compositions are yours from theirs?

**Answer:**
The ideal way to estimate the high *P-T* fluid composition requires knowledge of partition coefficients for Si, Na, K and Al between granitic fluids and melts. Those data are currently lacking and Anderson and Burham's solubility data (as well as Wohlers et al., 2010) are the closest composition we found to describe the exchange between the haplogranite melt and the aqueous fluid.
In general, our system involving peralkaline haplogranite glass is expected to contain more alkalis than the albite-$H_2O$ system. Yet, as mentioned above, the amount of dissolved Si and Na in the high P-T aqueous fluid remains low enough to prevent Br-Na complexation. Thus, the uncertainties in our fluid composition calculations are not expected to significantly affect Br partitioning behaviour.

**Comment 3:** Please could you display the Br concentrations in both phases as well as the D values calculated?
**Answer:**
Bromine concentrations were not directly determined in this study because the $D_{Br}$ values were directly calculated from the fluorescence signal after correction for the density and effective transmission of each phase. Accurate back calculations of the bromine concentrations from the partition coefficients require estimations of the volume of melt and fluid in the high P-T chamber, which are difficult to estimate since 1) the volume of hydrous melt is different from that of glass initially loaded in the cell and 2) the 2D visual observation do not enable to take into account changes in the thickness of the Re gasket with increasing *P-T*.
Nevertheless, to better address the referee's comment, we recalculate Br concentrations in the high P-T fluids and melts from the partition coefficients, assuming that the fluid:melt volumetric ratio was similar to the initial fluid:glass ratio. We consequently provide in the text **(Lines 339-341)** the range of concentrations of Br in the melts at the investigated P-T condition and report Br concentrations in the coexisting fluid and melt in Run1 on Figure 2. However, as those numbers are only indicative and have an unknown uncertainty, we prefer not to report them in Table 2. It should be stressed that these uncertainties are almost cancelled in equation (1) and D values are calculated with better accuracy (errors <10%).

**Comment 4:** Table 2- References such as Wohlers et al., 2012 are missing from the reference list.

**Answer:**
The missing reference was added to the reference list.

**Comment 5:** Partitioning was determined via the intensities of Br in the fluid and the melt (equation 1), with Figure 2 showing Br Ka intensity maps which are utilised to calculate Br concentrations. In this figure, the Au displays that its intensity is similar, if not greater than that of Br rich phases this could be a function of the gating used during analysis, as Br Kα is 11.9, and Au Lß is 11.5 which might explain its relative intensity on par with the measured Br. This begs the question: what gating size was utilised in this analysis, and did it have any effect on the overall Br concentrations recorded?

**Answer:**
It should be noted that Br fluorescence spectra were always collected far away from the Au chip, so that no Au signal was detected in the spectra. The Au $L_{\beta}$ was only observed in the spectra when analysis were taken ~ 5 μm away from the Au chip (horizontal beam size 8 μm) thus demonstrating the excellent beam shape and resolution, as well as the lack of secondary excitation that could compromise the quantitative analysis. **A sentence addressing this issue has been added to the revised manuscript (Lines 224-229).**

**Comment 6:** Br concentration was determined in the starting composition mixtures by various techniques. Noting the limitations of measuring Br via EPMA due to the overlap with Al (lines 104-106). However, by calculating the overlap of Br using the Lb line one can accurately measure Br efficiently; this can be used to check your RBS data and to be consistent between all measurements

**Answer:**
We thank the reviewer for this comment and will consider this option in future work.

**Comment 7:** Partitioning of Br is not measured in the same composition as those studied for speciation, making any comparison challenging.

**Answer:**
As Referee 1, Referee 2 argued that comparing the partitioning and speciation data may be challenging due to the different glass/melt compositions investigated in the two different set of experiments. **This comment has been addressed in the Answers to Referee 1 and a** note has been added in **Lines 479-482** to clarify this point and emphasize the similarities between the haplogranite and NS2 systems.

**Comments 8 and 9:** Once plotted all D values, versus P or T, shows little relationship and a lot of dispersion. Which makes it difficult in understanding how the partitioning of Br may vary along a subduction zone. Also, plotting D values in a density plot against P and T show no clear trends. Can this be discussed?
Can further discussion go towards how your results depict a change in Br partitioning with subduction?

**Answer:**
**Explanations for the lack of P-density-composition relationship have been provided to referee 1.**
In the present manuscript, the fluid-melt partition coefficients are only used to discuss the capacity of fluids and melts to incorporate Br (and Cl, by extension) and carry it inside and outside the subducting slab. Our conclusion is that both fluid and melt can be efficient media to mobilize Br and transfer it to the mantle wedge. We now further underline this point by reporting an average $D_{Br}$ value in the

abstract **(Lines 20-21).** We also point to the fact that a previous study by Bureau et al. (2010) also did not display any obvious P dependency at P > 0.2 GPa **(Lines 327-329).**

**Comment 10:** Figure 3 – the label at the bottom of the graph is not correct – this shows that Br preferentially is partitioned into the fluid. Br should have partitioned into both phases, but preferentially into the fluid.

**Answer:**
The point has been taken and the misleading labels were changed to '*Br partitions preferentially into the fluid/melt*'.

**Comment 11:** Were any standards measured at the synchrotron under the conditions tested, or are all the results based on those referenced in the paper such as Ferlat et al 2002 etc? If so a plot or a discussion of how good the fit to each speciation might be useful.

**Answer:**
A NaBr powder was analysed as the structural standard. This sample however yielded a noisy spectrum, probably due to some issues with the preparation of the pellet. Thus, we used the 3 wt% NaBr solutions (room conditions) and added the extra fit for the Ferlat et al. 2002, to ensure that our fitting procedure also enabled us to reproduce previous EXAFS analyses.

**Additional comments:**
- The title has been updated to specify the nature of fluids (ie., aqueous) and melts (ie., silicate).
- Supplementary materials have been incorporated into the main manuscript text to better provide the reader with as many details as possible and to better address the reviewers' concerns **(Lines 103-108; 115-118; 121-130).**
- A set of dashed lines was added to Figure 5 to underline the shift of EXAFS oscillations with change in composition. The caption of the figure was changed accordingly.

---

## Author Response (AR2)

**Topical Editor (N. Malaspina) comments:**

Dear authors,
the manuscript has been much improved even if one of the two reviewers, after a second run of revisions, expressed some concerns particularly regarding uncertainties on calculations. Please consider these further comments to complete the revision process.

**Answer:**
Dear Editor,

We wish to thank you for your decision.
In order to take into account the reviewer's concern about the underestimation of pressure and compositional uncertainties on the calculation of $D^{f/m}_{Br}$, we now report all values with a $2\sigma$ deviation that takes into account a 10% error on pressure determination and the analytical uncertainties on the intensity ratios. Figure 3 has also been updated so as to better visually display the uncertainty on $D^{f/m}_{Br}$ values and the relationship to density **(Fig. 3B)**. We also mention that the dispersion of the partition coefficients may arise from such uncertainties **(Lines 322-325).**

The reviewer's comments are answered in details as follows:

**Referee 1 additional comments:**

*I read carefully the answers to the two review reports and the new version of the manuscript. A lot of efforts have been done that have greatly improved the scientific quality of the manuscript. However, I believe that there are still a few unanswered questions. This is mostly due to the fact that some explanations provided by the authors are not satisfactory or are missing. These points must be clarified.*
*My answers are inserted in the following "comment/answers" text.*

**Comment 1:**

*The results about partitioning are equivalent to those of a previous published study by another group (nothing new) and the discussion about one low pressure result (at 0.2 GPa) is not convincing (see specific comments below). The results about bromine speciation in high pressure fluids are new and they deserve to be published, unfortunately the partitioning of Br is not measured for the same chemical system (haplogranite, HPG) than its speciation ($Na_2Si_2O_5$, NS2), which makes any comparison difficult. Therefore I would recommend to delete the part about partitioning, or at least to provide convincing explanation (see specific comments), and to focus on the speciation results.*

*Answers to comment 1:*

*1) While the in-situ Br partitioning experiments are not the first of their kind, they provide a unique opportunity for cross-checking experimental reproducibility and thus, we believe they deserve to be included in the manuscript. References to previous work by Bureau can be found throughout the manuscript and the favourable comparison between their and our studies further supports the reliability of the in-situ measurements. It should be stressed that in-situ measurements as those reported here and in rare previous work are extremely challenging, but the only reliable way to assess element partitioning and speciation at extreme P-T conditions, and thus any new data would be a valuable contribution to the field. Therefore, we prefer to keep the partitioning experiments as part of the current manuscript.*

- **Reviewer's answer**
**OK**

*2) Partitioning experiments involved haplogranite melts (Si, Al, Na, K), while Br speciation in melts could only be determined for sodium disilicate ($Na_2Si_2O_5$) due to insufficient Br concentrations in the haplogranite melt (400-2000 ppm Br). Yet, both XANES and EXAFS analyses (Figures 4 and 5; Table 3) show that Br local environment is very similar in the haplogranite and NS2 glasses. Thus, it can be expected that Br incorporation mechanism in both melts is similar and controlled by the presence of alkalis, either Na or K, and that all peralkaline silicate melts will have affinity to incorporate high amounts of Br under high P-T conditions. A note has been added in Lines 479-482 to clarify this point and emphasize the similarities between the haplogranite and NS2 systems.*

- **Reviewer's answer**
**It has been experimentally demonstrated that the solubility of Br in silicate melts is highly dependent of the composition of the silicate ($SiO_2$, Al/alkalis ratio, see Bureau Metrich, GCA, 2003), it is not convincing to claim that the speciation of Br should be the same for haplogranite and for NS2. Results obtained on glasses (Figure 4) cannot be used to predict the speciation of Br in melts.**

- **Authors comment:**
**The study from Bureau and Metrich indeed shows a decrease in solubility with increasing $SiO_2$ and Al/alkalis. Thus, both the lower $SiO_2$ and extreme Na concentrations in the NS2 should favor Br incorporation in this melt.**

Yet, higher concentration of Br may not only be related to Br speciation. In Louvel et al. (2020), for instance, it is shown that XAS spectra collected on natural basalt, andesite and rhyodacite glasses containing different amounts of Br share a lot of similarities, while they differ significantly from those from haplogranite glass. These differences are attributed to Br being surrounded solely by alkalis (Na and/or K) in the haplogranite glass, while alkaline earth ($Ca^{2+}$) may also be present in natural basalt, andesite and rhyodacite glasses. The higher solubility of Br in alkali-rich and more depolymerized glasses (e.g., basalt vs. rhyodacite) may thus only arise from the higher availability of alkali (and alkaline earth) cations, rather than actual differences in the incorporation mechanism of Br in the glass structure.

We agree with the reviewer that results obtained on glasses may not be fully representative of the high P-T melt speciation. This is actually what our EXAFS analysis suggests for NS2. However, as Br speciation is similar in hydrous NS2 and haplogranite glasses, it appears sensible to consider that increased P-T conditions (and $H_2O$ concentrations) will not modify the affinity of Br for alkalis and that Br in high P-T hydrous haplogranite melt will be surrounded by Na (and/or K) and oxygen/water. Thus, we wish to keep the sentence added on lines 480-483 to underline that similarities between NS2 and haplogranite glasses allow us to *anticipate* (and not *assert*) that Br will be found in alkali-dominated environment in the haplogranite melt.

*3) We believe that underlining the differences between our low P DBr (4.8 at 800 °C, 0.2 GPa) and those of Bureau et al. and Cadoux et al (17.5-20.2 at 900 °C, 0.2 GPa) is of relevance to this study to highlight that significant amounts of Br (and Cl) may be retained in degassed lavas, as reported in natural context. The discussion has been modified to highlight this point (Lines 356-364).*

- **Reviewer's answer:**
  I recognize that the authors have followed my recommendations about the discussion of $D^{f/m}$ at low pressure, and this is a good point, however what is proposed in the new discussion from line 354 to 361, is wrong. As explained in Balcone-Boissard et al., 2010, in some cases, the eruptive style allows a very fast decompression and the consequence is that degassing is not at equilibrium. In these cases, Br and Cl may be retained in significant amounts in the lavas, whereas if degassing is at equilibrium, they are totally washed out from the silicate melt. However, it cannot apply to the present experimental study, because the "experimental" degassing (i.e. decompression) is not fast enough, pressure was decreased slowly, from a step to another one in order to allow in situ measurements. Furthermore, in previous study, performed at the same conditions and with the same chemical composition it is shown that bromine is totally degassed. This is probably what the authors would have found if they would have analyzed the quenched samples.

  Authors comment:
  We agree and thank the reviewer for this comment: our assumption here was that if any halogens had previously been removed by deep fluid exsolution, significant amounts of Cl and Br would have remained in the

melts, and could then be retained upon fast (disequilibrium) degassing. That assumption was misleading and reference to the study of Balcone-Boissard et al. (2010) at the end of section 3 has hence been removed from the revised manuscript.

**Comment 2:**

*Br partition coefficients (D) are measured in situ for HPG system within the range 0.2 – 1.7 GPa and 592 – 840 °C: they are ranging from 4.1 to 15.3, they fall in the same range than those from Bureau et al., 2010, for similar conditions (0.66 –1.7 GPa, 590-890∘C, D from 2.18 to 9.2).*
*However, if one plots all results in a diagram D versus pressure, data exhibits a lot of dispersion and no real relationship, as it should be expected (i.e. an increase of D with decompression due to degassing). This is not discussed at all.*

**Answers to comment 2:**

*1) Referee 1 was right to point out the dispersion of the data and the apparent lack of correlation with P. A similar lack of pressure dependency (as well as density, or composition) has previously been reported for vapour/brine partitioning of some metals (Cu, Au, Ag). It was suggested that large differences in the speciation of these elements in both phases could be responsible for such a behaviour (Pokrovski et al., 2008). We thus suggest that the large differences in Br chemical and structural environment in between the coexisting fluid and melt phase could as well explain the scattered $D_{Br}$ in our study and the apparent lack of simple trends with P, density or dissolved silicate content, since the physical-chemical controls of Br in the two phases are very different due to the different speciation.*

- **Reviewer's answer**
  **If this would be right it should have been noticed in previous measurements of partitioning for the haplogranite-water-Br system. This is not the case. Br behavior cannot be compared to transition elements Cu, Au, Ag, having chemical properties far to those of halogens.**

- **Authors comment:**
  **Deviation from ideal behavior has also been underlined by studies on Cl, though for significantly higher Cl concentrations (Webster et al., 2018). We however believe that the current behavior of Br is yet too poorly constrained to discard such possibility. Only few studies are available, mostly at P < 200 MPa (Bureau et al., 2000; Cadoux et al., 2018) and the combination of Bureau et al. (2010) and our study only produced 20+ datapoints, with potential uncertainties on pressure determination inherent to *in-situ* technics in both studies. Explanations for the scatter in $D_{Br}$ are now detailed on lines 320-337. Comparison to Cl and the need for new data are also underlined in Figure 3B and lines 337-339 and 362-365.**

*2) We would also like to point out that with a single exception at 0.2 GPa, all our data were recorded at P > 0.5 GPa, where the existing studies have also shown that DBr values do not change significantly with pressure and exhibit a similar degree of scattering, between 1 and 10 (Bureau et al., 2010).*

- **Reviewer's answer:**
  **The major problem is the extreme dispersion of the partitioning coefficients with respect to pressure that is still not explained or even discussed by the authors: we expect a decrease of D with pressure, which is logical as close to total miscibility the D should be equal to unity.**

- **Authors comment:**
  **We agree with the reviewer and thus modified slightly the discussion of P-T-composition effect on lines 320 to 337 to 1) take into account the reviewer's concerns about pressure uncertainties and 2) provide other potential explanations for the dispersion of some of the data. We further underline that available data at high P-T may yet be too scarce to fully comprehend Br partitioning behavior under high P-T conditions.**

*3) We also note that additional discrepancies may arise from uncertainties on the estimation of the fluid composition, which will affect the calculated $D_{Br}$. We, for instance, recognize that the large $D_{Br}$ value obtained at 1.7 GPa is clearly off the trend, probably due to the fact that the fluid composition was calculated using the albite solubility data of Wohlers et al. (2011) instead of Anderson and Burnham (1983), to take into account the higher P conditions in this experiment.*

- **Reviewer's answer:**
  **Br concentration in all phases and Partition Coefficients can be calculated by using the PyMCA soft (see additional comment). This would provide an answer to that question.**

- **Authors comment:**
  **In this contribution, we chose not to rely on 'classical' SXRF quantification methods as they would have required:**
  **1) calibration of signal intensity, for aqueous fluids but supercritical fluids and hydrous melts, which have widely different properties from the aqueous fluids. The timeframe of synchrotron beamtime sadly do not enable to accommodate such time-consuming procedure when the scope of the experiments is to assess many different fluid and melt compositions;**
  **2) careful control of the sample thickness, which becomes difficult for long runs conducted under extreme temperature conditions in the HDAC, as the Re gasket accommodates P-T changes in a non-controllable manner.**
  **Therefore, we instead proposed the intensity ratio method, where corrections applied for the effect of different composition of fluid and melts (and density and absorption) are all discussed in the text.**

*4) We have added an additional discussion in the revised manuscript, both in the Results section (Lines 316-342) and in the Methods section, where we provide additional details about uncertainties on fluid and melt compositions and how they translate to the DBr (Lines 179-186 and 240-258). We also agree with this referee that the discussion of the temperature effect in a single experimental run was of weak relevance in terms of partitioning behaviour, and hence we have removed it from the revised manuscript.*

- **Reviewer's answer:**
  **OK**

*5) The conclusions drawn from our partitioning experiments, however, remain unchanged: we confirm that although Br preferentially partitions into the aqueous fluid over silicate melt, high amounts of Br can yet be incorporated in hydrous granitic melts. To strengthen this argument, we*

*added an estimation of Br concentration range in the high P-T melts of this study, calculated using the in-situ DBr and initial phase proportions (Lines 339-340).*

- **Reviewer's answer:**
  **It is not new that the amounts of Br in haplogranitic melts may be high at high pressure, see solubility data, but this is not what you have measured here. It would be useful to explain how you calculate the Br content of the melt, and why you have not used this to calculate the Br contents of the aqueous fluid and the partition coefficients.**

- **Authors comment:**
  **Our initial comment to the reviewer and the sentence added to the text on lines 341-343 (current version) explained that the Br concentrations in the melt were back-calculated from the $D_{Br}$ and initial fluid:glass ratio of the experiments.**

_**Comment 3:**_

_About the same value of D is obtained at 0.9, 0.8, 0.65, 0.2 GPa, respectively 4.4, 4.2, 4.1, and 4.8. High values of D are obtained at high investigated pressure (15 at 0.9 GPa and 9.7 at 1.7 GPa) where unity would be expected due to imminent total miscibility. Why?_
_Such a discrepancy may be due to the pressure determination. The authors use the diffraction of gold and its equation of state. However, it is well known that gold diffraction is not a good tool to for low pressure determination, as an example see Heinz and Jeanloz, JAP, 1983, where the first measurements are performed at 4.42GPa. Furthermore, in a recent intercomparison of the use of EOS for pressure determination (including Au), Ye et al., JGR, 2017, it is conclude that at high temperature, accuracy cannot be better than 1 GPa, from 2.5 GPa up to 140. For that reason, the discussion from line 278 to 294 is not relevant and should be suppressed._

_Answers to comment 3:_

_1) The choice of gold as the in-situ pressure calibrant was motivated by its chemical stability in high P-T fluids and melts. Although we agree that the absolute accuracy of this method may not be better than 1 GPa (Ye et al., 2017), the actual relative precision is much better and is likely to be within 10% of the P value, as reported in Louvel et al. (2014). Moreover, the unit cell volume of Au displayed systematic changes as a function of increasing P-T in the HDAC, thus demonstrating that Au is sensitive to relatively small pressure changes during the run, and special care was taken in the HDAC alignment/centering to preserve the sample-detector distance to ensure the reliability of the unit cell volume variations. Therefore, we are confident that the relative pressure variations during the run are captured by the Au pressure calibrant._
_To further support the appropriate pressure determination, we emphasize that the phase relations, including miscibility, in the haplogranite-$H_2O$ system are within the P-T range reported for other alkali silicate systems (e.g. Paillat et al., 1992; Stalder et al., 2000)._

- **Reviewer's answer:**
  **This is not convincing. The authors cannot recognize in a first sentence that "the absolute accuracy of this method may not be better than 1 GPa" and then write, "the actual relative precision is much better and is likely to be within 10% of the P value". Which would mean an order of magnitude less than what is well known from decades. If the authors are really sure about their accuracy they should provide a calibration. Otherwise it is not correct to calculate pressure with an accuracy of 0.1 GPa, and to use it to calculate the composition of the silicate content of the fluids (see comments/answers from/to reviewer 2). One explanation to justify the dispersion of $D^{f/m}$ data for a same pressure and the absence of relation between pressure and $D^{f/m}$, is that the pressure values may be so inaccurate that you cannot see it. A text must be added about pressure determination, with a realistic accuracy.**

- **Authors comment:**
  **Precision and accuracy are two difference things. In our experiments, the precision, which measures the reproducibility of the measurements, is well below 10% and we are convinced that the shifts in the lattice parameters of Au we are measuring are meaningful and the relative pressure changes are well constrained. We have never made any statement about the accuracy, which is the global error in the measurement, as it is difficult to estimate and requires taking into account instrument function and resolution that are not calibrated in most cases. Therefore, we do not have a calibration for the accuracy, which is certainly much worse than the precision that we are reporting.**

The different methods for pressure determination in the HDAC have been discussed at length in the past, and we believe there is currently no best approach available. Many studies rely on the liquid vapour homogenization curves for pure water to derive *in-situ* P (Shen et al., 1992). Those volumetric relationships cannot be extended to other fluid compositions involving high amounts of dissolved Cl or, as in our case, Si, Na and K. Solid pressure calibrants that are resistant to interaction with high P-T fluids are scarce. They include noble metals (Au, Pt) or minerals whose phase transition or vibrational properties have been previously calibrated up to the adequate P-T conditions (*e.g.,* quartz, zircon). Amongst those options, our only option at the time was to use Au, as quartz would have dissolved in the high P-T melts and a portable set-up was not available at SLS for *in-situ* Raman on zircon (Schmidt et al., 2013).

Additional mentions to potential uncertainties on the pressure determination has been added on lines 242-244, 254-257 and 322-324 to further take into account the reviewer's concerns. The manuscript now clearly states that the lack of relationship between pressure dependency and $D_{Br}$ may arise 1) from a larger uncertainty than estimated in the calculations (10%) and/or 2) from the significant effect of speciation.

*2) As mentioned above, additional discussion on the uncertainties of our calculations is now added both in the methods and result sections. Note that all fluid and melt properties (composition, density and effective transmission) were calculated assuming an uncertainty on pressure determination of 10% (Lines 183-186 and 245-246)*
.

- **Reviewer's answer:**
  **This may cause strong mistakes in your calculations.**

- **Authors comment:**
  **The reviewer's comment is answered above.**